# *Bacillus subtilis* TO-A suppresses the age-related decline in locomotion in *Caenorhabditis elegans*
Rika Inomata ⬡ ✉, Ryuichi Saito ⬡ , Diansheng Wang & Satoshi Shimazaki

Research on the role of probiotics in maintaining locomotor function in aged *Caenorhabditis elegans* is limited. Therefore, we investigated whether a probiotic strain, *Bacillus subtilis* TO-A (BSTOA), which extends the lifespan of worms can prevent age-related declines in the locomotor functions of aged worms. This study revealed that aged worms fed BSTOA (BSTOA_day13) maintained better locomotor function than those fed *Escherichia coli* OP50 (OP50_day13). BSTOA_day13 exhibited better activity than OP50_day13 in maintaining actin structure during age-related changes in body wall muscles. Notably, the expression levels of oxidative stress-related genes fluctuated in BSTOA_day13; these worms also displayed high survival rates based on the paraquat assay. An examination of the locomotor function of aged *gpx-7* mutants revealed that even worms fed BSTOA had a reduced thrashing rate. Therefore, consuming BSTOA may prevent the decline in age-related locomotor functions, such as symptoms of sarcopenia in humans, by inhibiting the decrease in *gpx-7* expression in aged worms.

The number and proportion of elderly individuals in the population have been increasing, particularly in high-income countries; however, low- and middle-income countries are also experiencing significant growth in this demographic. Between 2020 and 2050, the global population of those over 60 years of age is expected to double to 2.1 billion, and the population over 80 years of age is expected to triple to 426 million[1]. Experts expect that an extended healthy life expectancy, defined as the period during which individuals do not experience health-related limitations, will pose a considerable challenge for many as the elderly population ages. Sarcopenia, a skeletal muscle disorder commonly found in older adults and for which there is currently no effective treatment[2,3], is recognized as a significant factor influencing healthy life expectancy. Furthermore, the process of muscle mass loss typically begins at approximately 25–30 years of age and continues throughout life[4]. Therefore, it is crucial to develop innovative technologies to prevent and/or treat sarcopenia and reduce the associated risks.

Probiotics are defined by the Food and Agriculture Organization of the United Nations and the World Health Organization as "live microorganisms that, when administered in adequate amounts, confer beneficial health effects on the host"[5]. They are known to enhance bowel movements and maintain the homeostasis of digestive and immune functions[6], and it is anticipated that they may contribute to the extension of healthy life expectancy. A systematic review and meta-analysis by Shokri-Mashhadi et al. suggest that probiotic supplementation is important for the prevention of sarcopenia, functioning by improving muscle strength and enhancing the

gut microbiota diversity, and that long-term use is effective in preventing age-related muscle dysfunction. Supplementation with specific bacteria has the potential to alleviate the decline in locomotion associated with aging[7]. *Bacillus subtilis* TO-A (BSTOA) is one type of bacterium included in probiotic formulations, and studies report that it is effective against constipation, diarrhea, irritable bowel syndrome, enteritis, and other infectious diseases, with its history of use spanning over half a century[8–12]. Recently, studies have also reported its effects on lifespan extension in *Caenorhabditis elegans*, but further research is needed to clarify the specific characteristics of its anti-aging effects[13].

*Caenorhabditis elegans* is widely used in studies on developmental biology, genetics, and molecular biology[14]. Its advantages as a model organism, including ease of manipulation of growth conditions, a short lifespan, and other beneficial traits[15], make it a valuable tool in applied research, such as for screening processes and studies on aging and longevity. Owing to its loss of muscle integrity, reduced sarcomere density, and impaired directional sensing with age, leading to motor dysfunction and abnormal phenotypes, this organism is considered valuable for understanding the pathophysiology of sarcopenia[16,17]. Some researchers have demonstrated that microorganisms or probiotics, including *B. subtilis* can extend the lifespan and improve locomotion in these worms[18]. For example, researchers suggest that the nitric oxide and/or biofilms, produced by *B. subtilis*, mediate the lifespan-extending effects on worms[19,20]. Additionally, Koyuncu et al. reported that knockdown of the *eps-8* gene, for which

Research Division, TOA Biopharma Co., Ltd., Gunma, Tatebayashi-shi, Japan. ✉e-mail: r.inomata@toabio.co.jp

expression is upregulated with aging in worms, extends the lifespan and prevents muscle structure instability[21]. Based on these findings and having previously observed that worms fed BSTOA exhibit a less apparent decline in movement, we hypothesized that the ingestion of *B. subtilis*, known for its life-extending properties, could help prevent destabilization of the muscle structure. To date, there is limited research on the role of microorganisms or probiotics in maintaining locomotor function in aged worms. We believe that assessing not only the lifespan extension in worms but also their status (behavior, muscle structure, stress resistance, etc.) during aging could lead to novel insights into the effects of probiotic supplementation on healthy lifespan extension in worms and the underlying mechanisms. Therefore, in this study, we investigated the effect of BSTOA on sarcopenia-like disease signs in aged worms as an indicator of an extended healthy lifespan, aiming to elucidate the mechanisms underlying the prevention and/or treatment of sarcopenia using BSTOA.

## Results

### Effect of maintaining locomotor function in aged worms fed BSTOA

Age-related muscle aging was estimated using locomotion test parameters, such as the average defecation cycle length (s), pumping frequency (times min$^{-1}$), thrashing rate (times min$^{-1}$), and moving velocity (mm min$^{-1}$). Hatched *C. elegans* were fed *Escherichia coli* OP50 (OP50) for 2 days, and the synchronized worms were designated as young adults (OP50_day1). Young adults were continuously fed OP50 or BSTOA for 12 days, and these were defined as aged worms (OP50_day13 or BSTOA_day13, respectively). Both the thrashing rate (Fig. 1a) and pumping frequency (Fig. 1b and Supplementary Movie 1a) were significantly increased, while the average defecation cycle length (Fig. 1c) was notably shortened, in BSTOA_day13 compared to those in OP50_day13. However, there was no difference in the moving velocity between BSTOA_day13 and OP50_day1, although both were considerably higher than that of OP50_day13 (Fig. 1d and Supplementary Movie 1b). These results suggested that BSTOA ingestion suppresses the decline in locomotor function and the progression of aging in aged worms.

Next, to assess changes in the structures of the worm muscles involved in the locomotion tests mentioned previously herein, we performed actin staining to examine the structural changes in the body wall muscles, which are ideal tissues for observing the effects of aging and degeneration owing to their lack of cell-regeneration capacity in these worms[22]. The proportions of intact actin filaments in the body wall muscles in OP50_day1, BSTOA_day13, and OP50_day13 were 76.0%, 26.7%, and 83.3%, respectively (Fig. 1e). The micrographs in Fig. 1f show phalloidin-stained actin in the body wall muscles of worms under different bacterial feeding conditions, specifically around the posterior arm of the gonad. These results demonstrate that the body wall muscle structure of aged worms fed BSTOA is comparable, if not superior, to that of young adults. Therefore, one possible mechanism underlying the suppressive effects of BSTOA ingestion on the decline in locomotor function and the concomitant anti-aging effect could be associated with body wall muscle homeostasis.

### Characteristic gene expression in aged worms fed BSTOA

Preventing and treating sarcopenia, which has various causes[23,24], can be better understood through a comprehensive analysis of the changes in gene expression caused by aging. Therefore, we investigated variations in the expression of genes related to aging and locomotion in aged worms fed BSTOA. The target genes were well-known genes related to worm aging and locomotion[21,25–33], chosen to investigate differences in gene expression between aged worms fed BSTOA and OP50, using that in young adults as the reference value of 1.0, by performing quantitative PCR (Fig. 2).

Compared to those in OP50_day1, the expression levels of four genes, namely *pmk-1*, *eps-8*, *rac-2*, and *asm-3*, and two genes, namely *gst-38* and *asm-3*, increased by more than two-fold in OP50_day13 and BSTOA_day13, respectively. Meanwhile, those of six genes, namely *unc-54*, *ifb-2*, *gst-4*, T20H4.5, *sucg-1* and *acdh-13*, and two genes, namely *unc-54* and *ifb-2*,

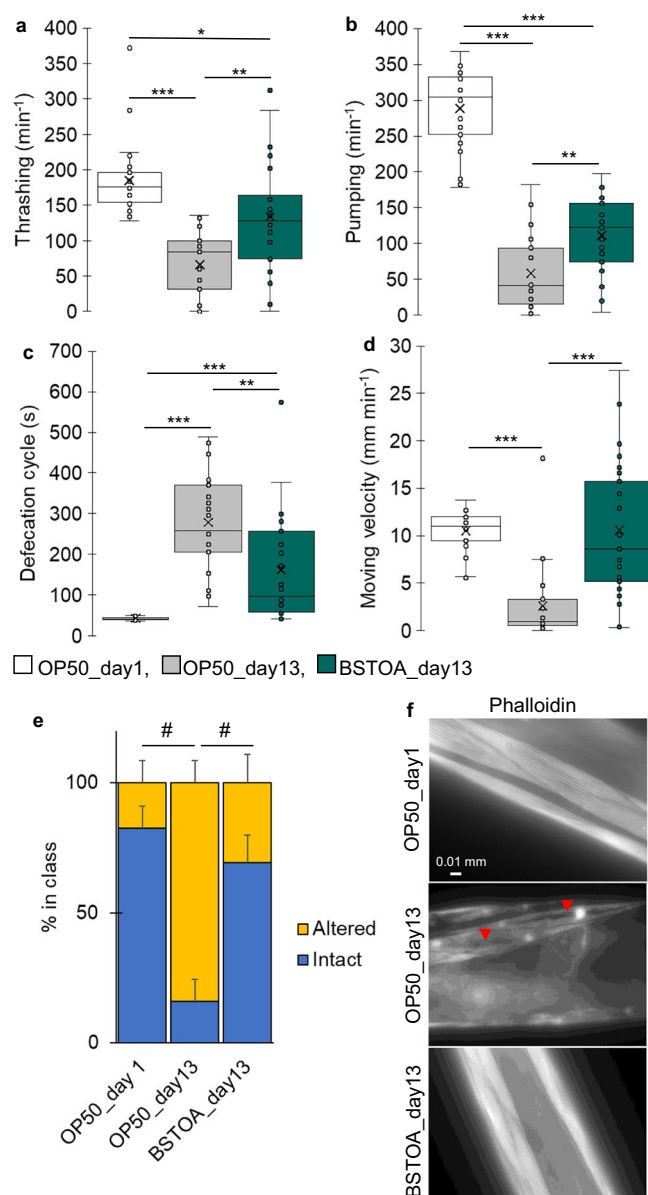

**Fig. 1 | Locomotion tests to assess locomotor function and the status of body wall muscles in aged worms under each bacterial feeding condition.** In young adults fed OP50 (OP50_day1, white) and aged worms fed OP50 (OP50_day13, gray) or BSTOA (BSTOA_day13, green), **a** thrashing rates, **b** pharyngeal pumping frequency, **c** defecation cycle lengths, and **d** moving velocities were expressed by box plots, from three independent experiments; each dot represents one nematode ($n = 25$ per condition). The statistical analysis was performed with the Kruskal–Wallis test and post-hoc Steel–Dwass test, and *p* values were calculated based on a two-tailed, unpaired Student's *t*-test (*$p < 0.05$, **$p < 0.01$, ***$p < 0.001$) (**a–d**). **e** Percentage of individuals with intact or disrupted actin in the body wall muscles under each condition. Statistical significance, #, was determined based on Tukey's multiple comparison test for proportions ($\alpha = 0.05$). OP50_day1, $n = 45$, OP50_day13, $n = 50$, and BSTOA_day13, $n = 50$. **f** Representative micrographs of stained actin with phalloidin in the body wall muscles of worms under different bacterial feeding conditions around the posterior arm of the gonad. The red triangle indicates the altered area in the body wall muscle. Results were independently replicated at least three times.

were decreased to less than half of OP50_day1 levels in OP50_day13 and BSTOA_day13, respectively (Fig. 2a). Furthermore, expression levels of four genes involved in oxidative stress, *unc-54*, *gst-4*, *gst-38*, and *asm-3*[34–38], were upregulated by 2.8-fold, 3.1-fold, 4.7-fold, and 5.4-fold, respectively in BSTOA_day13 compared to those in OP50_day13 (Fig. 2b).

**Fig. 2 | Quantitative PCR analysis of genes related to aging and locomotion in aged worms.**
**a** Comparison of the delta-delta Ct values between OP50_day13 (gray) and BSTOA_day13 (green). Individual values are indicated as dots, and error bars indicate the standard error of the mean (SEM) from three replicates. **b** A heatmap indicating expression levels in BSTOA_day13 vs. OP50_day13. Blue and red indicate downregulation and upregulation, respectively. *p*-values were calculated based on a two-tailed, unpaired Student's *t*-test (*$p < 0.05$, **$p < 0.01$, ***$p < 0.001$).

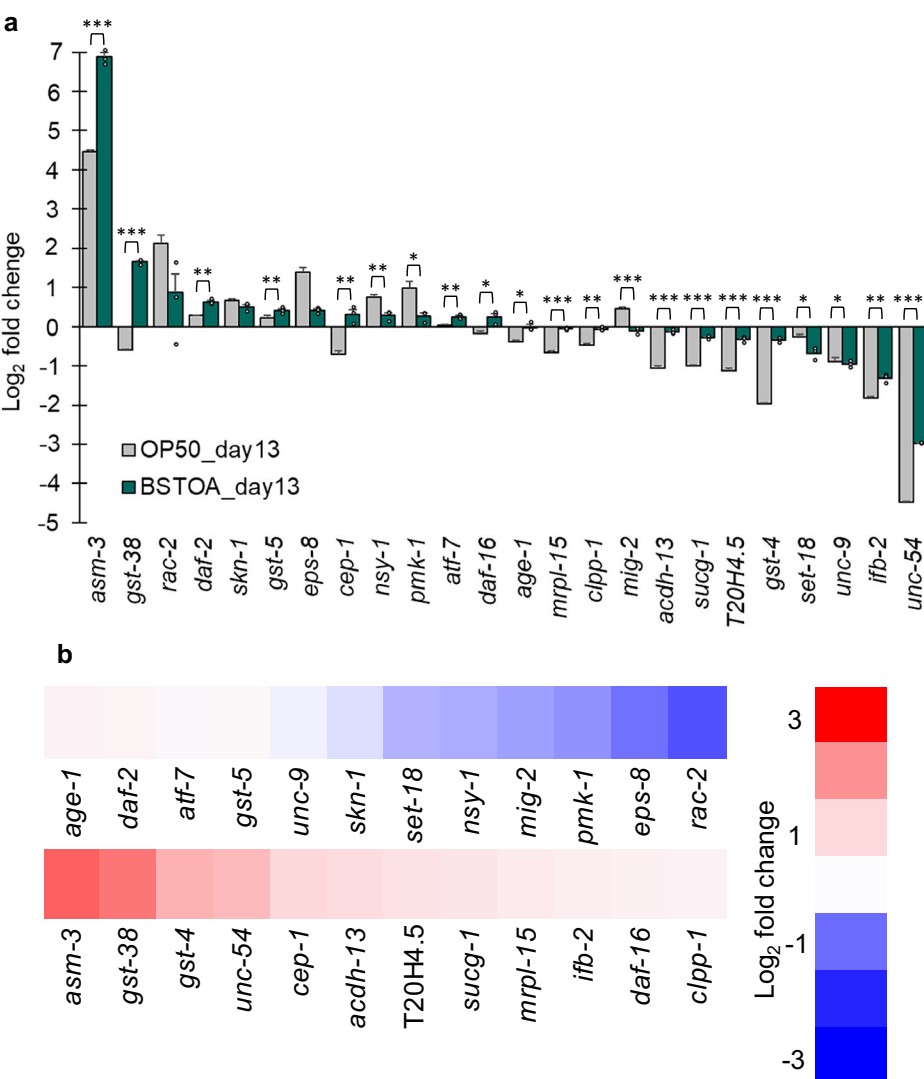

## Locomotor function and oxidative stress in aged worms: BSTOA vs. BS168

It was expected that BSTOA would influence the responses to oxidative stress associated with aging, thereby helping to suppress the decline in locomotor function due to senescence. Figure 2 shows that the expression of genes involved in oxidative stress tended to fluctuate in BSTOA_day13 compared to that in OP50_day13. To further test our hypothesis and determine whether the effect in aged worms is unique to BSTOA, *B. subtilis* 168 (BS168), known to prolong the lifespan of worms, was selected for a comparison[39].

First, locomotor function was evaluated based on the thrashing rate in OP50_day1, OP50_day13, BSTOA_day13, and BS168_day13. As mentioned previously herein, locomotor function was maintained in BSTOA_day13, whereas BS168_day13 showed a decline in locomotor function comparable to that in OP50_day13 (Fig. 3a). Next, it was hypothesized that the increased expression of genes associated with oxidative stress observed in BSTOA_day13 (Fig. 2) would influence the resistance of aged worms to oxidative stress. Consequently, a paraquat assay was performed. BSTOA_day13 exhibited a significantly higher survival rate, of ~87%, in the paraquat assay with a high concentration of paraquat, whereas BS168_day13 demonstrated a significantly lower survival rate, of ~10%, similar to that in OP50_day13 (Fig. 3b). Aged worms fed BSTOA exhibited exceptionally high oxidative stress resistance, as BSTOA_day13 had a higher survival rate than OP50_day1 under paraquat conditions (Table 1). These

results suggest that strain BS168 affects the longevity of worms but not their locomotor function or oxidative stress resistance. Functional differences were clearly observed among *B. subtilis* strains, indicating that the suppression of locomotor dysfunction and enhancement of oxidative stress resistance would be a characteristic of BSTOA.

To characterize the expression of genes related to oxidative stress resistance in BSTOA_day13, for which changes typically occur as the decline in locomotor function is suppressed (Fig. 3a), gene expression levels in BSTOA_day13 were compared with those in OP50_day13 and BS168_day13, with those in OP50_day1 used as the reference value of 1.0 (Fig. 3c). Oxidative stress-related genes that exhibited less than a 1/2-fold or more than a 2-fold change in expression on BSTOA_day13 or BS168_day13, compared with that in OP50_day13, included six genes, *ctl-1*, *ctl-2*, *ctl-3*, *gpx-4*, *gpx-7*, and *gst-4*, and seven genes, *ctl-1*, *ctl-2*, *ctl-3*, *gpx-5*, *gst-4*, *gst-5*, and *gst-38*, respectively. In addition, among the oxidative stress-related genes that showed more than a 2-fold increase or decrease in expression levels in BSTOA_day13, compared with those in OP50_day13, the expression levels of *gpx-4* and *gpx-7* exhibited similar fluctuating patterns between OP50_day1 and BSTOA_day13, whereas they displayed opposite patterns between BS168_day13 and OP50_day13. Specifically, *gpx-4* expression increased by more than 2-fold in OP50_day13 and BS168_day13 compared with that in OP50_day1, whereas no significant change was observed in BSTOA_day13. In contrast, *gpx-7* expression levels in OP50_day13 and BS168_day13 were approximately one-eighth those in

**Fig. 3 | Effects of *Bacillus subtilis* strains on locomotor function, oxidative stress resistance, and gene expression in aged worms. a** Thrashing rates as a measure of locomotor function. **b** Survival rates, based on a paraquat assay, as a measure of oxidative stress resistance. **c** qPCR analysis of the expression of oxidative stress-related genes. The *p*-values were determined based on Tukey's test (\**p* < 0.05, \*\**p* < 0.01). Individual values are represented as dots. **d** The heatmap indicates the degree of gene expression in BSTOA_day13 and BS168_day13 compared to that in OP50_day13. Blue and red indicate downregulation and upregulation, respectively. Error bars indicate the SEM from three replicates.

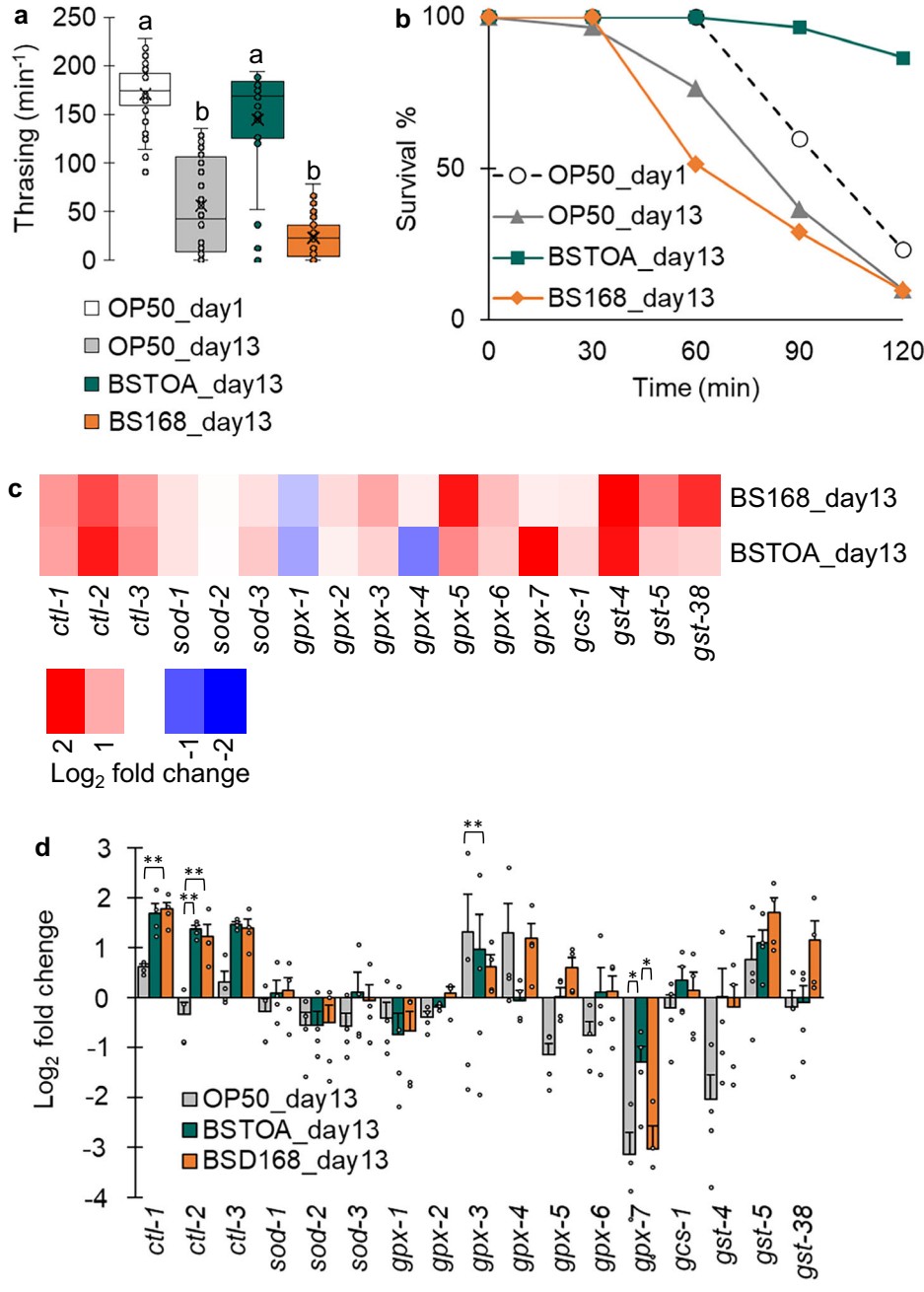

OP50_day1. However, in aged worms, the age-associated decrease in *gpx-7* expression was suppressed in BSTOA_day13, with ~2-fold higher expression compared with that in OP50_day13 and BS168_day13. Therefore, the BSTOA diet appears to maintain more youthful levels of *gpx-7* expression, curtailing the age-related decline in expression (Fig. 3c, d).

### BSTOA influences *gpx-7* expression to preserve aged worm mobility

Based on the results shown in Fig. 3c and d, characteristic changes were observed in the oxidative stress-related genes *gpx-4* and *gpx-7* in BSTOA_day13. To ascertain whether these genes are involved in maintaining locomotor function and oxidative stress tolerance in aged worms fed BSTOA, we performed locomotion tests and paraquat assays using *gpx-4* and *gpx-7* mutants. Regarding thrashing rates, both BSTOA_day13 and OP50_day13 *gpx-4* or *gpx-7* mutants showed a decline in locomotor function, with no significant differences between these aged worms (Fig. 4a and

Supplementary Fig. 1a). However, in wild-type worms, BSTOA_day13 showed no decline in locomotor function, which was maintained at almost the same level as that in OP50_day1, whereas a decline in locomotor function was observed in OP50_day13 and BS168_day13 (Fig. 3a). These results suggest that BSTOA specifically affects the expression of *gpx-7* during the thrashing motion involving the body wall muscles, preventing a decline in locomotor function and promoting anti-aging effects.

In contrast, regarding the mean defecation time in *gpx-7* mutants, BSTOA_day13 had a significantly shorter mean defecation time than OP50_day13. Nevertheless, in *gpx-4* mutants, there was no significant difference between BSTOA_day13 and OP50_day13 (Fig. 4b and Supplementary Fig. 1b). These results suggest that after BSTOA ingestion, *gpx-4*, but not *gpx-7* expression might be associated the defecation function of aged worms, which is regulated by multiple factors such as the muscles around the anus, body wall muscles, and nervous system[40,41]. The survival rate in the paraquat assay (Fig. 4c and Supplementary Fig. 1c) and pumping frequency

(Fig. 4d and Supplementary Fig. 1d) were both significantly higher in BSTOA_day13 than in OP50_day13 for the *gpx-4* and *gpx-7* mutants. These results were similar to those of the wild-type strain (Fig. 1), suggesting that the effects of BSTOA ingestion on oxidative stress resistance and pumping behavior are not related to the presence or absence of *gpx-4* or *gpx-7* gene expression, but rather involve other genes.

### Table 1 | Multiple comparisons of survival rates in paraquat assays under different bacterial feeding conditions

| Condition | | | Log rank test Bonferroni *p*-value |
|---|---|---|---|
| OP50_day1 | vs. | OP50_day13 | |
| OP50_day1 | vs. | BSTOA_day13 | *** |
| OP50_day1 | vs. | BS168_day13 | * |
| OP50_day13 | vs. | BSTOA_day13 | *** |
| OP50_day13 | vs. | BS168_day13 | |
| BSTOA_day13 | vs. | BS168_day13 | *** |

## Discussion

Sarcopenia is diagnosed through a comprehensive assessment of muscle mass and function based on measurements and imaging, muscle strength using grip strength assessments, and physical ability based on walking speed[2]. Therefore, in this study, the effect of BSTOA on *C. elegans*, aside from lifespan extension, was examined using a combination of methods, including muscle quality, determined based on stained images of the muscle fibers, and muscle functions assessed through exercise tests. Consequently, the continuous ingestion of BSTOA by the worms resulted in maintenance of the intact muscle structure and locomotor function, suggesting its anti-aging effects on aged worms (Figs. 1 and 3a, b). Notably, BS168, which is known to have a lifespan-extending effect[13], did not influence maintenance of the locomotor function in aged worms, indicating that the lifespan-expanding effects of probiotics are not correlated with the maintenance of locomotor function. Therefore, this is the first study to propose that BSTOA influences both lifespan extension and the preservation of locomotor function by suppressing the age-related decline in *gpx-7* gene expression.

To elucidate how BSTOA ingestion extends the lifespan and maintains motor function, we analyzed gene expression in aged worms fed BSTOA. Here, the expression of several representative genes was analyzed via qPCR

**Fig. 4 | Effect of BSTOA on locomotor function and oxidative stress resistance in aged *gpx-4* and *gpx-7* mutants. a** The thrashing rate, **b** evacuation time, **c** survival rate, based on a paraquat assay after 2 h, and **d** pumping frequency were measured using aged *gpx-4* and *gpx-7* mutants. Individual values are shown as dots, and these experiments were performed at least twice independently. Error bars indicate the SEM. Different letters indicated significantly different results from each other at *p* < 0.05.

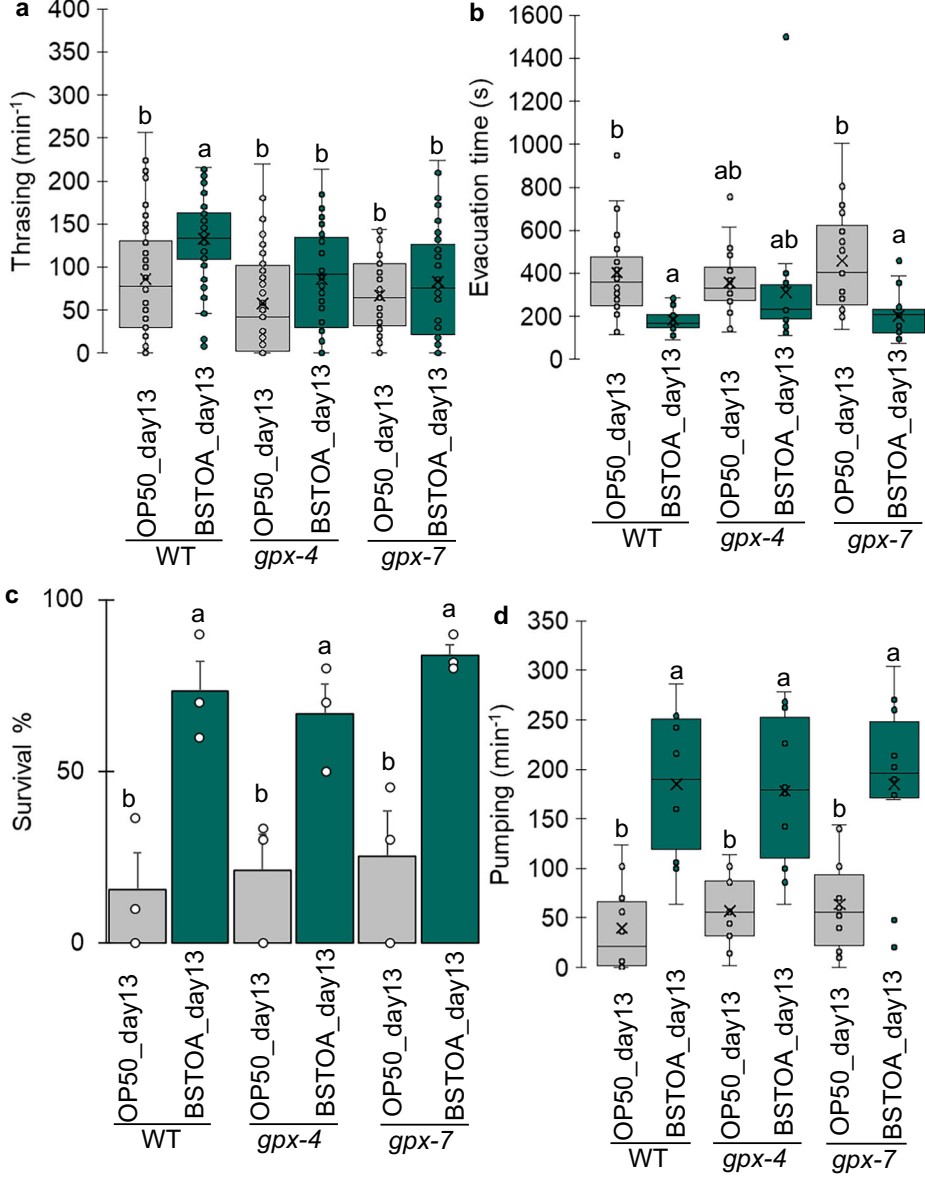

among the many genes related to aging and movement. The expression levels of genes associated with oxidative stress were upregulated more than two-fold (Fig. 2). Therefore, we shifted our primary focus to oxidative stress resistance. We assessed this using a paraquat assay, in which the survival rates of worms exposed to a high concentration of paraquat were examined. Notably, the survival rates of BS168_day13, OP50_day1, and OP50_day13 declined to ~10% of control levels at 2 h, whereas BSTOA_day13 maintained a high survival rate of ~80% at the same time point. Unexpectedly, the survival rate of BSTOA_day13 was significantly higher than that of OP50_day1 (Fig. 3b). Moreover, whereas worms were reared with OP50 until day 13, the worms showed evident physical deterioration due to aging, and the anti-aging effects of BSTOA were different from those of OP50 on day 13. For example, generally, the body lengths of worms stop growing, and they exhibit intestinal atrophy with aging[42,43]. Nevertheless, the body length of BSTOA_day13 was significantly increased compared to that of OP50_day13, and the intestinal atrophy observed in BSTOA_day13 was less pronounced compared to that in OP50_day13 (Supplementary Fig. 2). This phenomenon observed in BSTOA_day13 may explain why the survival rate of this group, with higher concentrations of paraquat, was greater than that of OP50_day1, based on the paraquat assay conducted. Although not examined in this study, it is also possible that the anti-aging effects in each tissue contribute to the maintenance of locomotor function and the paraquat resistance.

Consequently, these findings suggest that *gpx-4* and *gpx-7* are related to aging and oxidative stress resistance in worms, as they regulate glutathione peroxidase (GPx) activity, which is predicted to participate in cellular oxidative detoxification. Further, these genes are human *GPX3* (or *GPX5*) and *GPX4* orthologs[44,45], with GPx converting hydrogen peroxide to water and reducing organic hydroperoxides to alcohol forms[46] using glutathione (GSH) as an electron donor, thus protecting cells from oxidative damage. Not only GPx but also antioxidants and enzymes, such as superoxide dismutase and catalase, function together to eliminate reactive oxygen species. However, despite predictions of enhanced oxidant detoxification, *gpx-4* and *gpx-7* mutants appear to have no discernible difference in paraquat resistance. Sakamoto et al. showed that *gpx-1*, *gpx-2*, *gpx-6*, and *gpx-7* are not essential for growth, reproduction, physiological rates, or protection against exogenous oxidative stress in *C. elegans*[45], with quadruple *gpx* mutants exhibiting no significant effects in response to exogenous oxidative stress, highlighting the difficulty of detecting phenotypes based on single *gpx* mutants. In addition, Ko et al. revealed that both *gpx-4* and *gpx-7* influence neuronal axon regeneration by regulating lipid oxidation, which was shown to influence axonal damage repair[47]. Maintaining optimal oxidative levels improves axon fusion efficiency, implying that the corresponding mutants might suffer from impaired neuronal axon repair and reduced locomotion functionality. We suggest that there is a possibility that BSTOA supplementation might help regulate the oxidative balance, potentially inducing anti-aging effects.

In this study, body wall muscles were primarily analyzed based on stained images and locomotion tests, particularly thrashing motility. Nonetheless, as per the locomotion tests (Fig. 1a–d), it is likely that the cardiac-like pharyngeal muscles contract during pumping motility, while the anal sphincter, the intestinal smooth muscle, and skeletal muscle-like body wall muscles are involved in defecation. Comprehensively considering the results of these tests, it is suggested that the anti-aging effects of BSTOA extend not only to skeletal muscle-like body wall muscles but also to all types throughout the body. Moreover, we investigated day-13 *gpx-4* and *gpx-7* mutant adults fed OP50 or BSTOA (Supplementary Fig. 3). Even in the *gpx-4* and *gpx-7* mutant strains, the body wall muscle structure of aged nematodes fed BSTOA remained relatively more stable than that in worms fed OP50. However, the body wall muscle stability in aged nematodes of the *gpx-4* mutant strain was lower than that in the N2 and *gpx-7* strains. Since *gpx-4* is predicted to be localized to the extracellular space, its closer proximity to the body wall muscle may have exerted a more pronounced effect on the muscle structure than that observed in the *gpx-7*-mutant strain.

This study illustrates how the muscle developed in young adulthood is maintained and functions, as the number of non-gonadal cells in adult worms (959 somatic cells in hermaphrodites) remains unchanged[15,48]. This suggests that the continuous ingestion of BSTOA could inhibit the degradation of the muscle structure, decline in locomotor function, and acquisition of oxidative stress resistance in aged worms. In the future, we plan to investigate not only the effects on maintaining the structure and function of existing muscle cells but also the influence on muscle regeneration. By comprehensively verifying the effects of BSTOA ingestion on muscle remodeling, we aim to explore the potential use of probiotics for the prevention and/or treatment of human sarcopenia, for which there is still no effective treatment in the era of a 100-year lifespan.

## Methods

### Bacterial strains and culture conditions

Bacterial strains used in the present study were as follows: *E. coli* OP50 (OP50) was acquired from the Caenorhabditis Genetic Center (CGC), *B. subtilis* TO-A (BSTOA) was used as a manufacturing product, and *B. subtilis* 168 (BS168) was purchased from the Biological Resource Center, NITE (NBRC). OP50 was grown in 5 mL of Luria-Bertani (LB) broth at 37 °C, shaking at 100 rpm overnight after cultivation on LB agar at 37 °C. BSTOA and the BS168 were grown in 5 mL of trypticase soy (TS) broth, prepared from TS agar (Eiken, Tokyo, Japan) after removing the agar, with shaking, at 37 °C overnight. These cultures were diluted in LB or TS broth to an optical density at 600 nm of 0.5. To prepare a bacterial lawn as feed for worms, 100 μL of the bacterial suspension or 10 mg of bacteria washed with and suspended in M9 was dropped onto a 60 mm nematode growth medium (NGM) plate or a modified NGM (without peptone, mNGM) plate, respectively. The NGM plates were incubated at 37 °C under aerobic conditions.

### *Caenorhabditis elegans* strains and growth conditions

These strains were as follows: *C. elegans* N2 Bristol as the wild type, provided by CGC; *gpx-4* (tm2111), and *gpx-7* (tm2166), provided by the National Bioresource Project (NBRP). For culture and maintenance, we followed the methods of previous reports[13]. Worms were cultured on an NGM plate with a lawn of OP50 at 20 °C. To synchronize the worms, the hermaphrodite adults were allowed to lay eggs for approximately 4 h at 20 °C on the NGM plates with the bacterial lawn; after the hermaphrodite adults were removed, the hatched worms were cultured for another 2 days to become young adult worms, which were designated as day-1 adults. Then, the day-1 adult worms were transferred to fresh plates containing OP50, BSTOA, or BS168 every 2 days, and culture was continued for another 12 days to obtain mature worms, designated as respective aged day-13 adult worms. Each experiment was conducted as an independent biological replicate, using different cohorts of nematodes that were separately prepared for each experiment[49].

### Measurement of defecation

The defecation time was measured using a previously reported method[13]. Defecation behavior was evaluated by measuring the time between the expulsion of intestinal contents using a stereomicroscope (SZ2-ET, OLYMPUS, Tokyo, Japan). The mean defecation time or cycle was calculated either once or for five consecutive defecation cycles. Each experiment was performed in triplicate, with at least five worms per experiment.

### Pharyngeal pumping frequency

The worms were acclimated to room temperature (RT), and the pharyngeal muscles of each group were imaged on the plate under each feeding condition using a stereomicroscope with a camera (WRAYCAM-EL310, WRAYMER, Osaka, Japan), independently based on two or three separate experiments. The videos were trimmed to 30 s using video-editing software (Windows Movie Maker ver. 9.9.9.5) to count the number of movements. Then, the pharyngeal pumping frequency was doubled to calculate the pumping rate per minute.

 

## Thrashing motility assay

A *C. elegans* thrashing assay was performed in liquid medium, as a well-known method to assess motility[50]. We picked several worms from the plate and transferred them to a well containing 200 μL of M9 buffer in a 96-well plate. The worms were video-recorded after stabilization for 30 s. The procedure, from trimming the videos to calculating the counts, was similar to that used for the pharyngeal pumping frequency described previously here. A "thrash" was defined as bending the body to the outermost corner in one direction and returning it to the initial position. Three independent experiments were performed, using ~10 worms per condition.

## Moving velocity

Young adult and aged worms from the OP50_day1, OP50_day13, and BSTOA_day13 groups were transferred individually to new plates with the corresponding dietary conditions. Soon thereafter, one worm was video-recorded for ~30 s. We measured the distance crawled over a 30 s period and calculated the velocity, using image processing software (ImageJ 1.53k, NIH, Bethesda, MD, USA).

## Actin staining

This method was performed as described previously[21]. We collected 10 worms per a PCR tube containing M9 buffer and washed them twice with M9 buffer and once with PBS-0.2T, which contained PBS (pH 7.2) and 0.2% Tween 20. After the worms were fixed with a 4% paraformaldehyde solution for 30 min at RT, they were washed a third time with PBS-0.2T. Subsequently, as a pretreatment for dye penetration, worms were treated overnight at RT with a β-ME solution (120 mM Tris–HCl [pH 6.8], 5% β-mercaptoethanol, 1% Triton X-100, and 2% Tween 20). Worms were stained with the Phalloidin-iFluor™ 555 Conjugate (1:100, Cayman Chemical, Ann Arbor, MI, USA) for 4 h after washing three times with a blocking buffer (PBS-0.2T and 1% BSA). The images (Fig. 1f) are representative examples of worms from each strain. Because the fluorescence was quickly quenched, it was difficult to photograph all the worms evaluated. The status of body wall muscles from head to tail was assessed by two observers using an inverted fluorescence microscope (ECLIPSE Ti-S, Nikon, Tokyo, Japan) equipped with a ×60, 1.40 numerical aperture (NA) oil-immersion objective. An individual was judged to be intact if both observers assessed the absence of any disruptions in actin bundles, such as ruptures in the darkly appearing body wall muscle or pointed, dashed, or fragmented actin filaments. Meanwhile, the individual was judged to be altered if the worm showed any disruption of actin bundles. The image was processed using ImageJ software.

## RNA purification and quantitative RT-PCR for *C. elegans*

One hundred worms from each group were collected in the lysis solution[51], consisting of 1% Triton X-100, 1% Tween 20, 0.5 mM EDTA, 5 mM Tris–HCl, pH 8.0, 16% RNAsecure™ RNase Inactivation Reagent (Thermo Fisher Scientific Inc., WA, USA), and 1 mg/mL proteinase K, and were immediately frozen in liquid nitrogen. To extract RNA, these solutions were heated at 65 °C for 15 min and then deactivated at 85 °C for 1 min. After performing ethanol precipitation on these solutions, total RNA was purified using NucleoSpin® RNA (Takara Bio Inc., Shiga, Japan)[52]. The cDNA was synthesized using the PrimeScript™ RT reagent Kit with gDNA Eraser (TaKaRa Bio), and RT-PCR was performed using TB Green® Premix Ex Taq™ II (TaKaRa Bio) and the Bio-Rad CFX96 real-time PCR System (Bio-Rad Laboratories, Inc., CA, USA). The expression of each gene was measured in triplicate, and its Ct value was averaged across three wells; the obtained data were then normalized to the average Ct value of two (*tba-1* and *cdc-42* [Fig. 2]) or five genes (*tba-1*, *cdc-42*, *eif-3*, *ama-1*, and *pmp-3* [Fig. 3]) as the ref. 53 within each experiment. The designed primers are listed in Supplementary Table 1. We used the delta-delta threshold cycle (Ct) method ($2^{-\Delta\Delta Ct}$) to express the difference between groups. Delta Ct values were generated by subtracting the average Ct of the reference genes from the Ct of each gene of interest. Then, the ΔΔCt was derived from the comparison with a control sample. The ΔΔCt results were plotted on graphs using a logarithmic axis (Figs. 2a and 3c).

## Paraquat assay

Several worms per well were treated with either 0 or 240 mM paraquat, with ~10 worms per test, and observed every 30 min for up to 2 h, or until 2 h had passed. A dead worm was defined by the absence of any reaction despite tapping the head and tail with a platinum wire under a stereomicroscope[54].

## Percentage of the intestines in the body

The areas of the body and intestine of nematodes photographed under a bright field microscope (SZ2-ET, OLYMPUS) were measured using ImageJ by outlining the contours of each. In total, 40 worms were used per condition, with three replicates.

## Statistical analyses

Statistical analyses were performed using R (version 4.4.4) with the Kruskal–Wallis test and post-hoc Steel–Dwass test or one-way analysis of variance with Tukey's test, Student's *t*-tests, or the Chi-square test. The survival percentage in the paraquat assay tracking time course was evaluated using the log-rank test in OASIS 2[55] or Tukey's test. Statistical significance was defined as $p < 0.05$. These values are annotated as $*p < 0.05$, $**p < 0.01$, or $***p < 0.001$, or letters a, b, c, and d indicate significant differences between groups ($p < 0.05$).

## Reporting summary

Further information on research design is available in the Nature Portfolio Reporting Summary linked to this article.

## Data availability

All data are available in the main text and/or the supplementary materials. The source data behind the graphs in the paper can be found in Supplementary Data.

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

## Acknowledgements
We would like to thank the CGC for supplying the *C. elegans* and *E. coli strains* and the NBRP for providing the mutants. We would also like to thank Editage (www.editage.jp) for English language editing. Finally, we are grateful to the research members of TOA Biopharma Co., Ltd., for their fruitful discussions.

## Author contributions
Conceptualization: R.I. and R.S. Methodology: R.I. and R.S. Investigation: R.I. and R.S. Visualization: R.I. Supervision: S.S. D.W. Writing–original draft: R.I. and R.S. Writing–review & editing: R.I., R.S., D.W., and S.S.

## Competing interests
The authors declare no competing interests.
