## [Transparent Peer Review file · Communications Biology]

Bacillus subtilis TO-A suppresses the age-related decline in locomotion in *Caenorhabditis elegans*

Corresponding Author: Dr Rika Inomata

Version 0:

Reviewer comments:

Reviewer #1

(Remarks to the Author)

The manuscript entitled: "Bacillus subtilis TO-A suppresses age-related decline in locomotion of *Caenorhabditis elegans*". The manuscript will be reviewed after major revision. Specific comments are listed below:

Comments:

1. Numerous studies have reported the beneficial effects of *Bacillus subtilis* on organismal health using various in vivo models. Why, then, do the authors choose to revisit this topic? Is there a specific reason driving this study?
2. Several probiotic strains have been shown to improve locomotion behaviour in worms at later stages of life. How, then, can it be claimed that this is the first study to suggest that this particular strain enhances both longevity and the maintenance of locomotor function?
3. Locomotor function in worms fed BSTOA was studied on day 13. The longevity of *C. elegans* was up to 20 days. But the locomotion was studied on 13th day. How the authors conclude that BSTOA suppresses age-related decline in locomotion. Why the study was not conducted on later stages.
4. The thrashing behaviour of the worms on the 13th day was reported to be close to 300 times per minute. Is it feasible for thrashing to reach this frequency at such an advanced stage? Additionally, the reported movement velocity also seems unusually high.
5. Include video recordings of locomotion behaviour and pharyngeal pumping on the 13th day in supplementary file for reference.
6. Is it possible to conduct the study using the GFP-tagged MYO-3 marker strain RW1596 (*myo-3::GFP*) to assess muscle fiber organization at later stages after treatment?
7. Statistical significance should be added to Fig. 2.
8. Include the gene expression analysis results, along with the fold changes, in the results section.
9. Include the oxidative stress tolerance of *C. elegans* on the 13th day as a percentage in the results section.
10. Elaborate further on the connection between longevity, GPX-7, and locomotion behaviour in the results and discussion sections.
11. How many worms were used in a single experiment to assess oxidative stress?
12. Make sure that all species names are italicized and check their formatting throughout the manuscript.
13. Explain the role of *gpx-7* gene in the introduction section.
14. Statistically significant changes should be marked in Figures 3b and 3c.
15. Why was the oxidative stress assay not conducted using juglone?

Reviewer #2

(Remarks to the Author)

The manuscript by Inomata et al. found that *Bacillus subtilis* TO-A (BSTOA) suppresses age-related locomotor decline and actin tissue injury. They have claimed in this study that the BSTOA strain specifically suppresses locomotor, but in all experiments they compared it to *E. coli* OP50 and/or to *Bacillus subtilis* strain 168, which showed a longevity effect. To support the hypothesis that BSTOA is specific, a comparison should be made with *Bacillus subtilis* strains that do not show longevity effects and/or do not affect locomotor function. Although BSTOA showed a resistance effect to paraquat, the

oxidative stress-related genes *gpx-4* and *gpx-7*, which contributed to the suppression of locomotor function decline, were not related to paraquat resistance. Therefore, the detailed mechanisms and relationships between locomotor function and antioxidant stress are unclear in this study. It is necessary to obtain data to determine the mechanism you are proposing, not by using effector genes as previously reported, but by using transcription factors related to oxidative stress and using techniques for analyzing comprehensive genes. The current results are only limited to the phenomenon that TOA suppresses locomotor function decline and actin damage compared to *E. coli* OP, thus lacking in impact. In the future, if it is possible to clarify part of the mechanism involved in these phenomenon and to prove that BSTOA is specific, it will have a significant impact. Therefore, it is insufficient and inappropriate to accept the manuscript in its current state.

Major concerns:

- Line 110 and 111

If you hypothesize that the oxidative stress associated with aging is suppressed, nematodes fed *B. subtilis* that do not extend lifespan should be used as a control group.

- Line 127-129

It is difficult to conclude at this time that the effects of BSTOA on suppression of locomotion and resistance to oxidative stress are specific to BSTOA. It is already reported that *B. subtilis* is resistance to hydrogen peroxide, a type of oxidative stress (<https://doi.org/10.1093/jambio/lxad082>). It could be possible to argue that it is specific by comparing it to various strains such as this strain.

- Line 173-176

From the flow of the story, it is hypothesized that the decline in exercise with aging is related to oxidative stress, and what kind of oxidative stress genes and which ones are involved have been investigated. However, it was finally described that *gpx-4* and *gpx-7* were not directly involved in oxidative stress in BSTOA, although they were involved in locomotion and defecation in BSTOA. On the other hand, why was *gpx* genes not involved in oxidative stress in paraquat, even though the above discusses that *gpx* genes is involved in oxidative stress? It is contradictory.

Minor points:

- Line 49

Reference 14 is not a paper on BSTOA.

- Line 47-49

Did the BSTOA-fed nematodes grow and sexually mature at the same rate as the *E. coli* OP-fed nematodes? If these markers were also delayed, it is difficult to say that it is anti-aging.

Not only should it be compared to *E. coli* OP, but it should also be compared to strains that do not extend lifespan with *B. subtilis*.

- Line 94-107, Fig. 2

How did you choose which genes related to aging and locomotion for qPCR?

- Line 137-140 and 144-150

The sentences are similar. It is better to describe them together.

- Line 169 and 170

Please cite the reference.

- Line 212 and 213, Supplementary Fig. 1

The method of measuring body length and intestinal atrophy is not provided in the methods section. If intestinal atrophy was assessed by the area value of the intestinal region, then it would simply be BSTOA accumulates in the intestine more than *E. coli* OP, so the area value is greater.

- Method for moving velocity

If the movement speed were examined in a medium coated with each bacterium, then it is possible that they avoided BSTOA. It is possible that they are running away from it, and it is difficult to claim that the high migration speed suppresses the decrease in locomotion.

- Line 303

Sequence information of the primers used in qPCR should be noted.

- Line 313 and 314

Why did you use 2 genes in Figure 2 and 5 genes in Figure 3 as housekeeping?

Please indicate the reason for the different number of genes in the housekeeping.

- Fig. 1e and 1f

Can you use this actin-based muscle staining to investigate the involvement of *gpx-4* and *gpx-7*?

- Fig. 4a

To determine if *gpx-4* and/or *gpx-7* are involved in the suppression of locomotion reduction in BSTOA, it would be better to compare *E. coli* OP in wild type fed BSTOA.

•Fig. 4b

Is there some previous reports that defecation time increases with aging? Please cite the references for this.

Muscle mass (muscle site) regulated by gpx-4 and gpx-7 are different, so is it possible that bending is regulated by gpx-7 and defecation is regulated by gpx-4 may be involved?

Reviewer #3

(Remarks to the Author)

Version 1:

Reviewer comments:

Reviewer #1

(Remarks to the Author)

The revised manuscript titled "Bacillus subtilis TO-A suppresses age-related decline in locomotion of *Caenorhabditis elegans*" offers a well-executed and insightful investigation into the protective effects of *B. subtilis* TO-A against age-related physiological decline in *C. elegans*. The authors have carefully addressed all previous reviewer comments, resulting in substantial improvements to the manuscript's clarity, scientific rigor, and overall quality. The presented data convincingly demonstrate that *B. subtilis* TO-A alleviates the age-associated deterioration in locomotor function in *C. elegans*. Revisions to the methodology section have enhanced transparency, and the figures and legends are now more informative and accessible. Furthermore, the statistical analyses are appropriate and well-applied. Given the comprehensive nature of the revisions and the robustness of the findings, I find the manuscript to be suitable for publication and recommend acceptance in its current form.

Reviewer #2

(Remarks to the Author)

Manuscript Number: COMMSBIO-24-7847A

Bacillus subtilis TO-A suppresses age-related decline in locomotion of *Caenorhabditis elegans* by Inomata et al.

The reviewer pleased that the authors have resolved all the questions. However, based on your response, there are three points that should be clarified.

1. Regarding gpx-4 and gpx-7

The involvement of these genes as you describe in your response letter should be validated using mutants lacking both or transgenics with reduced expression by RNAi.

2. If the gene examined by qPCR is the major gene reported so far, the literature should be cited where the gene is mentioned.

3. About the housekeeping used in qPCR

I understood that 5 genes were used as housekeeping to minimize the effect in aging. But how did you correct the expression levels for these five genes? Did you just simply graph the average of the 5 data? Please describe in detail how you calculated and graphed the data in the Methods section.

Reviewer #3

(Remarks to the Author)

Version 2:

Reviewer comments:

Reviewer #2

(Remarks to the Author)

This is an appropriate response and should be accepted.

Re: COMMSBIO-24-7847

Title: *Bacillus subtilis* TO-A suppresses age-related decline in locomotion of *Caenorhabditis elegans*

Authors: Rika Inomata, Ryuichi Saito, Diansheng Wang, and Satoshi Shimazaki

We would like to thank all the referees for their thoughtful and insightful evaluation of our manuscript. Following their valuable comments and suggestions, we have revised our manuscript by including new data and modifying the text. Our point-to-point responses and explanations to all the comments of the reviewers are as follows:

Response to Reviewer 1:

Thank you very much for your time and efforts in reviewing our paper.

This research explored the potential applications of a clinically applied probiotic strain of *Bacillus subtilis* TO-A (BSTOA) in new fields and their mechanisms of action, providing scientific evidence. Although life expectancy continues to increase in countries around the world, the rate of elderly people requiring nursing care continues to increase at a high rate, placing a heavy burden on families and society. We have pursued this research with the belief that if we can extend human healthy lifespan, particularly by targeting sarcopenia associated with the arrival of the 100-year-old era, it will have a significant impact on reducing healthcare and social insurance costs.

Your constructive suggestions and comments about the work were very helpful for improving our manuscript.

1. Numerous studies have reported the beneficial effects of *Bacillus subtilis* on organismal health using various in vivo models. Why, then, do the authors choose to revisit this topic? Is there a specific reason driving this study?

Thank you for your important question. As you mentioned, reports have shown the beneficial effects of *Bacillus subtilis* (BS) on organismal health using various in vivo models. However, unlike other BS, BSTOA is used as an ethical pharmaceutical for stool abnormalities in Japan. And we believe its significance lies in demonstrating the novel beneficial potential of BSTOA, especially for sarcopenia, under the 100-year life era. Furthermore, it is well-known that bacteria within the same species may exhibit different functionalities depending on the strain. While BSTOA has already shown lifespan-extending properties in another report, its efficacy in preserving locomotor function has remained unclear. Therefore, we considered it essential to present additional data to investigate this aspect in more detail.

2. Several probiotic strains have been shown to improve locomotion behaviour in worms at later stages of life. How, then, can it be claimed that this is the first study to suggest that this particular strain enhances both longevity and the maintenance of locomotor function?

Thank you very much for your comment. Several probiotic strains have indeed been reported to contribute to the maintenance of locomotor function or to extend lifespan in nematodes. However, this study is the first to demonstrate that BSTOA exerts both locomotor function maintenance and lifespan extension effects (Fig. 3). Accordingly, we have revised the text in Lines 197–199 as follows: “Therefore, this is the first study to report that BSTOA has effects on both lifespan extension and the maintenance of locomotor function through

suppressing the age-related decline in *gpx-7* gene expression.” We hope this revision clarifies the intended message.

3. Locomotor function in worms fed BSTOA was studied on day 13. The longevity of *C. elegans* was up to 20 days. But the locomotion was studied on 13th day. How the authors conclude that BSTOA suppresses age-related decline in locomotion. Why the study was not conducted on later stages.

As you pointed out, utilizing nematodes nearing the end of their lifespan could make the difference between nematodes ingesting OP50 and those ingesting BSTOA more pronounced, as nematodes consuming OP50 exhibit extremely sluggish movement, almost to the point of immobility. However, nematodes at this stage are difficult to collect in sufficient numbers, and their near immobility complicates the observation of movement to an extent suitable for quantitative evaluation. Therefore, in this study, we decided to use 13-day-old nematodes, a time point more conducive to movement-based assessment. Additionally, upon other reports on aging-related studies using nematodes, similar age groups are frequently employed, suggesting that the 13 days for the evaluation is appropriate.

4. The thrashing behaviour of the worms on the 13th day was reported to be close to 300 times per minute. Is it feasible for thrashing to reach this frequency at such an advanced stage? Additionally, the reported movement velocity also seems unusually high.

Thank you for your comment. Based on the staining images of actin fibers, similar results were observed between OP50_day1 and BSTOA_day13, and the fact that individuals exceeding 284 were observed in OP50_day1 (Fig. 1b), I think that it may be possible that nematodes equivalent to late-stage BSTOA_day13 could also reach close to 284. However, the average value for BSTOA_day13 was 134 ± 81.3 . Regarding locomotion speed, the value of BSTOA_day13 was 10.6 ± 7.31 mm/min, which is almost unchanged from that of OP50_day1 (10.5 ± 2.24 mm/min), but BSTOA_day13 showed a high degree of variability.

5. Include video recordings of locomotion behaviour and pharyngeal pumping on the 13th day in supplementary file for reference.

Thank you very much for your comment. We have provided representative video files of the movement behavior on day 13 and the pharyngeal pumping as supplementary files.

6. Is it possible to conduct the study using the GFP-tagged MYO-3 marker strain RW1596 (*myo-3::GFP*) to assess muscle fiber organization at later stages after treatment?

As you might know, RW1596 is a strain where the GFP-tagged *myo-3* is expressed in the *myo-3* mutant (st386), thereby rescuing the phenotype of *myo-3*. Additionally, since the *myo-3* mutant (st386) causes embryonic lethality, the *rol-6* mutant gene (su1006) has been introduced for maintenance and passage of the strain. Furthermore, RW1596 is not a strain with GFP-tagged *myo-3* inserted into the genome; thus, the overexpression of GFP-tagged *myo-3* or abnormal protein aggregation caused by the GFP tag may occur. Based on these considerations, we have opted for a method that involves staining and observing wild-type muscle fibers to ensure we only observe the effects of BSTOA intake.

We also thought that conducting research using RW1596 is feasible, and we plan to explore this approach in future research projects. Thank you for your valuable advice.

7. Statistical significance should be added to Fig. 2.

Thank you for your feedback. After conducting the statistical analysis, we have added the *p*-value to Fig. 2.

8. Include the gene expression analysis results, along with the fold changes, in the results section.

Thank you for your suggestion. The main text has been revised. Please refer to the revised main text in Lines 110–116 and 147–158.

9. Include the oxidative stress tolerance of *C. elegans* on the 13th day as a percentage in the results section.

Thank you for your feedback. I have updated the main text. Please refer to Lines 131 and 133 in the revised main text.

10. Elaborate further on the connection between longevity, GPX-7, and locomotion behaviour in the results and discussion sections.

Thank you for your suggestion. I have added it to the Discussion. Please refer to Lines 227–236 in the revised main text.

11. How many worms were used in a single experiment to assess oxidative stress?

Thank you for your question. We used 9 to 12 animals per group.

12. Make sure that all species names are italicized and check their formatting throughout the manuscript.

Thank you for your feedback. After confirmation, we have made the necessary corrections.

13. Explain the role of *gpx-7* gene the introduction section.

Thank you for your valuable comments. Additional information has been included in the Discussion section. Please refer to Lines 219–233 in the revised main text for details.

14. Statistically significant changes should be marked in Figures 3b and 3c.

Figure 3b could not present all combinations on the graph, so Table 1 shows statistically significant changes for all combinations.

Figure 3c has undergone statistical testing. Thank you for your comment.

15. Why was the oxidative stress assay not conducted using juglone?

Thank you for your question. According to the literature using the nematodes investigated, paraquat was employed for oxidative stress assays; thus, paraquat was used instead of juglone in this study. In the future, it may be possible to verify the effect of BSTOA using juglone as an alternative.

Response to Reviewer 2:

We sincerely thank you for your enthusiastic review of our manuscript and for your valuable comments and advice. Your constructive suggestions and comments about the work were very helpful for improving our manuscript.

This research explored the potential applications of a clinically applied probiotic strain of *Bacillus subtilis* TO-A (BSTOA) in new fields and their mechanisms of action, providing scientific evidence. Although life expectancy continues to increase in countries around the world, the rate of elderly people requiring nursing care continues to increase at a high rate, placing a heavy burden on families and society. We have pursued this research with the belief that if we can extend human healthy lifespan, particularly by targeting sarcopenia associated with the arrival of the 100-year-old era, it will have a significant impact on reducing healthcare and social insurance costs.

Major concerns:

• Line 110 and 111

If you hypothesize that the oxidative stress associated with aging is suppressed, nematodes fed *B. subtilis* that do not extend lifespan should be used as a control group.

Thank you for your suggestions. Indeed, comparing with *Bacillus subtilis* strains that do not extend the lifespan of nematodes would yield clearer results. However, despite literature research, we could not find any *B. subtilis* strains that suppress oxidative stress without prolonging lifespan, which led us to conduct comparative studies using BS168. (DOI: 10.1016/j.celrep.2019.12.078, DOI:10.1111/jam.15156)

The primary aim of this study is to explore the health-span-extending effects and mechanisms of action of probiotic strains, with a focus on preserving physical fitness and suppressing oxidative stress.

• Line 127-129

It is difficult to conclude at this time that the effects of BSTOA on suppression of locomotion and resistance to oxidative stress are specific to BSTOA. It is already reported that *B. subtilis* is resistance to hydrogen peroxide, a type of oxidative stress (<https://doi.org/10.1093/jambio/lxad082>). It could be possible to argue that it is specific by comparing it to various strains such as this strain.

As pointed out, it is currently difficult to conclude that the inhibitory effect on locomotor dysfunction and resistance to oxidative stress are specific to BSTOA. As shown in the revised main text Lines 137–139 "Functional differences were observed among *B. subtilis* strains, indicating that the suppression of locomotor dysfunction and enhancement of oxidative stress resistance would be characteristic of BSTOA." we have revised the text. Comparative studies with *B. subtilis* var. *natto* Miyagino strain (MI-OMU01), the target strain of the recommended literature, will be considered in future research. Thank you very much for your suggestion.

•Line 173-176

From the flow of the story, it is hypothesized that the decline in exercise with aging is related to oxidative stress, and what kind of oxidative stress genes and which ones are involved have been investigated. However, it was finally described that gpx-4 and gpx-7 were not directly involved in oxidative stress in BSTOA, although they were involved in locomotion and defecation in BSTOA. On the other hand, why was gpx genes not involved in oxidative stress in paraquat, even though the above discusses that gpx genes is involved in

oxidative stress? It is contradictory.

Thank you for your comment. I apologize for the confusion caused by the insufficient explanation. Based on the results of thrashing (Fig. 3a), survival rates in a paraquat assay (Fig. 3b), and qPCR (Fig. 3c), it is considered that the oxidative stress-related genes *gpx-4* and *gpx-7* are involved in the decline in locomotive ability and paraquat resistance. Therefore, we proceeded with experiments using *gpx-4* and *gpx-7* mutants. The results indicated "These results were similar to those of the wild-type strain (Fig. 1), suggesting that the effects of BSTOA ingestion on oxidative stress resistance and pumping behavior are not related to the presence or absence of *gpx-4* or *gpx-7* gene expression alone but to the involvement of other genes." (revised main text, Lines 182–185). However, the mutant nematode strains used in this study each have mutations in individual genes, and considering that the *gpx* family consists of multiple members, it is possible that the effects cannot be fully confirmed with single-gene mutations alone. Therefore, we reviewed the literature and discussed the findings as described in the revised main text (Lines 227–236). We hope that these revisions clarify the intent of our study.

Minor points:

- Line 49

Reference 14 is not a paper on BSTOA.

Thank you for your feedback. The text has been modified accordingly.

- Line 47-49

Did the BSTOA-fed nematodes grow and sexually mature at the same rate as the *E. coli* OP-fed nematodes? If these markers were also delayed, it is difficult to say that it is anti-aging.

Not only should it be compared to *E. coli* OP, but it should also be compared to strains that do not extend lifespan with *B. subtilis*.

As all nematodes were cultured on OP50 from the egg stage to adulthood, their sexual maturation is ensured without any issues. Regarding your comment on the comparison with *B. subtilis* that does not extend the lifespan of nematodes, as mentioned in response to your initial question, literature research did not yield any reports of *B. subtilis* strains that suppress oxidative stress without extending lifespan. Therefore, we conducted a comparative study with BS168. Furthermore, since this paper focuses not on lifespan extension effects but on the anti-aging effects in muscle, the issue you raised will be considered in future research.

- Line 94-107, Fig. 2

How did you choose which genes related to aging and locomotion for qPCR?

Thank you for your question. Based on a review of publications relating to aging and locomotion of worms, we selected genes that showed particularly significant changes in the data previously reported in the papers.

- Line 137-140 and 144-150

The sentences are similar. It is better to describe them together.

Thank you for your feedback. We have revised the manuscript. Please refer to the revised main text in Lines 147–158.

- Line 169 and 170

Please cite the reference.

Thank you for your valuable comment. The following reference has been added to the manuscript:

Thomas, J. H. Genetic analysis of defecation in *Caenorhabditis elegans*. *Genetics*, **124**(4), 855-872. (1990).

•Line 212 and 213, Supplementary Fig. 1

The method of measuring body length and intestinal atrophy is not provided in the methods section. If intestinal atrophy was assessed by the area value of the intestinal region, then it would simply be BSTOA accumulates in the intestine more than *E. coli* OP, so the area value is greater.

Thank you for your comment. A representative photograph has been added as Supplementary Fig. 1c, and the measurement method for body length and the paragraph on intestinal shrinkage titled "The percentage of the intestines in the body" have been included in the "Materials and Methods" section.

It is generally known that aging leads to observable phenomena such as the shrinkage of intestinal cells and the reduction of nematode body length (Herndon, L.A., Wolkow, C.A., Driscoll, M., and Hall, D.H. (2018). Introduction to Aging in *C. elegans*. In WormAtlas. doi:10.3908/wormatlas.8.4). As you pointed out, it is possible that BSTOA might accumulate more in the intestines of nematodes compared to OP50; however, we could not observe such a phenomenon in the microscopic analysis conducted in this study. Rather than your point raised, we consider that the influence of the nematode's body length and the degree of intestinal shrinkage is more significant. So, we have included supplementary data.

•Method for moving velocity

If the movement speed were examined in a medium coated with each bacterium, then it is possible that they avoided BSTOA. It is possible that they are running away from it, and it is difficult to claim that the high migration speed suppresses the decrease in locomotion.

Thank you for your comment. We consider that the probability of nematodes escaping from BSTOA is low. Based on our experience, nematodes grown on lactic acid bacterial strains often exhibit avoidance behaviors, but nematodes grown on BSTOA rarely appeared to escape to the walls during observation. Additionally, many nematodes remained within the lawn area, suggesting no apparent aversion to BSTOA. Further, the reason for investigating the movement speed of nematodes on media coated with each condition of bacteria was to minimize environmental changes, as transferring nematodes living on BSTOA to other bacterial surfaces might potentially alter their behavior. To substantiate locomotor activity, we not only measured movement speed but also analyzed the number of thrashing events per unit of time.

•Line 303

Sequence information of the primers used in qPCR should be noted.

Thank you for your comment. The sequence information of the primers used for qPCR has been added as Table 2.

•Line 313 and 314

Why did you use 2 genes in Figure 2 and 5 genes in Figure 3 as housekeeping?

Please indicate the reason for the different number of genes in the housekeeping.

In the first qPCR experiment aimed at estimating related genes, two typical housekeeping genes were used as references. Subsequently, to further narrow down candidate genes, five housekeeping genes were utilized in

the second qPCR experiment. Although widely used housekeeping genes are generally considered stable, it was hypothesized that their expression levels might fluctuate significantly during experiments involving aged nematodes. Thus, in experiments designed to refine the selection of genes, we took to minimize variability between experiments.

•Fig. 1e and 1f

Can you use this actin-based muscle staining to investigate the involvement of *gpx-4* and *gpx-7*?

Thank you for your suggestion. We carried out additional experiments and included the data in Supplementary Fig. 2.

•Fig. 4a

To determine if *gpx-4* and/or *gpx-7* are involved in the suppression of locomotion reduction in BSTOA, it would be better to compare *E. coli* OP in wild type fed BSTOA.

As you pointed out, we have added a comparison graph of BSTOA and OP50 in wild-type nematodes and revised Fig. 4a accordingly.

•Fig. 4b

Is there some previous reports that defecation time increases with aging? Please cite the references for this. Muscle mass (muscle site) regulated by *gpx-4* and *gpx-7* are different, so is it possible that bending is regulated by *gpx-7* and defecation is regulated by *gpx-4* may be involved?

Collins, J.J., et al. The measurement and analysis of age-related changes in *Caenorhabditis elegans* (December 7, 2007), WormBook, ed. The *C. elegans* Research Community, *WormBook*, doi/10.1895/wormbook.1.137.1, <http://www.wormbook.org>.

Thank you for your feedback. The following references have been cited; Collins, J.J., et al. The measurement and analysis of age-related changes in *Caenorhabditis elegans* (December 7, 2007), WormBook, ed. The *C. elegans* Research Community, *WormBook*, doi/10.1895/wormbook.1.137.1, <http://www.wormbook.org>.

In our test results, the effect of BSTOA intake on suppressing the age-related decline in thrashing motion was not observed in the *gpx-7* mutant strain, and although the effect of BSTOA intake on suppressing the age-related extension of defecation time in the *gpx-4* mutant strain was not significant, it appeared to be slightly weakened. As you pointed out, it is plausible that *gpx-7* is involved in regulating the suppression of muscle aging, while *gpx-4* may play a role in controlling muscle aging related to defecation. However, since the influence of the *gpx-4* mutant strain on defecation is relatively small, likely, many other genes not identified in this study might also be involved.

Response to Reviewer 3:

We are extremely grateful for your attentive review of our manuscript and for your sensitive, constructive, and valuable comments and advice. Your constructive comments were very helpful for improving our manuscript.

This research explored the potential applications of a clinically applied probiotic strain of *Bacillus subtilis* TO-A (BSTOA) in new fields and their mechanisms of action, providing scientific evidence. Although life expectancy continues to increase in countries around the world, the rate of the elderly requiring nursing care continues to increase at a high rate, placing a heavy burden on families and society. We have pursued this research with the belief that if we can extend human healthy lifespan, particularly by targeting sarcopenia associated with the arrival of the 100-year-old era, it will have a significant impact on reducing healthcare and social insurance costs.

Major concerns

1. My reading of the Methods and Results section did not leave me with a clear understanding of what the authors considered to constitute replicates in their experiments. Typically, the standard in the *C. elegans* field is that experiments are repeated at least three times with independent biological replicates, meaning with separate cohorts of animals that are derived from different synchronization events, achieved here by harvesting eggs laid within a 4 hour time window. If instead the authors repeated their experiments with different sets of animals that all came from the same cohort, then they have completed only technical replicates which would not be consistent with the field's practices. The authors should either clarify their methodology or generate additional datasets from independent biological replicates.

Thank you for your constructive comments. Each experiment was conducted as an independent biological replicate, using different cohorts of nematodes derived from distinct synchronized events (achieved by collecting eggs laid within 4 hours). We have added the following clarification in the Methods section (revised main text, Line 283–285): “Each experiment was conducted as an independent biological replicate, using different cohorts of nematodes that were separately prepared for each experiment.”

2. The authors present intriguing results to suggest that the BSTOA diet delays the age dependent deterioration of specific tissues in *C. elegans*, particularly body wall muscle and the intestine. To make these data more compelling, however, several revisions are needed and additional experiments are suggested, as described below. Further, the 2 authors should consider expanding their visual analysis of muscle tissue to substantiate some of their claims.

• **The authors need to clearly articulate the criteria that they used to categorize the actin bundles as either “altered” or “intact”.**

What were the specific features of “altered” actin bundles that distinguish them from “intact” bundles? Did the authors observe a range in the severity of the disorganization of the actin bundles as they examined groups of animals? If so, what were the minimum criteria that categorize them as “altered”?

Thank you for your thoughtful comments. Regarding the evaluation criteria, we have added the following explanation to the method section (revised main text, Lines 331–335): “An individual was judged to be intact

if both observers assessed the absence of any disruptions of actin bundles, such as ruptures in the darkly appearing body wall muscle or pointed, dashed, or fragmented actin filaments. Conversely, an individual was judged to be altered if the worm showed any disruption of actin bundles. The image was processed using the ImageJ software.”

- **In Fig 1e, the authors should include standard error bars** to indicate the range of values associated with each category observed across separate replicates of the experiment.

Thank you for your comment. Although we conducted the experiments multiple times, the vertical axis of the graph represented the proportion calculated based on the total number of nematodes. As per your suggestion, we have modified the graph to include error bars. Along with the graph revision, we have also updated the description of the figure accordingly. (Fig. 1e)

- The micrograph of actin filaments in body wall muscle of OP50_day1 animals is clear and at an appropriate level of magnification and brightness. Although the striations from actin bundles are apparent in the micrograph of BSTOA_day13 animals, some regions of the image are oversaturated, making a direct comparison to the OP50_day1 image difficult. **The authors should consider showing a different image of body wall muscle from BSTOA_day13 animals that is free of saturated pixels.**

Thank you very much for your constructive feedback. I have revised the manuscript based on your comments and replaced the image with an alternative one, as shown in Fig. 1f.

- **The micrograph of the body wall muscle in an OP50_day13 animal (Fig. 1f) does not clearly show the “altered” actin bundles and should be replaced with a new micrograph.** The image as shown is difficult to interpret because of the presence of the gonad and because the relative size of the body wall muscles depicted are smaller than those that appear in the other two micrographs. Was this image taken at the same magnification as was used to capture the other two micrographs or does the muscle structure as depicted here illustrate the effects of aging on the tissue? This image also appears to be overexposed with several supersaturated regions, similar to the micrograph of body wall muscles in BSTOA_day13 animals.

Thank you very much for your delicate and constructive evaluation. The images shown in Fig. 1f were all captured at the same magnification (60×). The fibers in OP50_day13 appear thinner, and the fluorescence is inherently weaker. As phalloidin stains actin in tissues other than the body-wall muscle, efforts to visualize the body-wall muscle led to stronger fluorescence signals in the intestine. Previous images with overexposure have been replaced following your recommendation to ensure higher visual clarity. Additionally, triangular markers have been incorporated to highlight areas identified as damaged for improved interpretation (Fig. 1f).

- In the Discussion section the authors state, “it is suggested that the anti-aging effects of BSTOA extend not only to...body wall muscles but also to all types throughout the body” (lines 220-222). **The authors should strongly consider expanding their analysis of muscle tissue architecture in BSTOA-fed animals to include pharyngeal muscle** since their behavioral assays indicate that BSTOA improves the rate of pharyngeal pumping in older animals. This could be accomplished by fluorescence microscopy. Including these data in the manuscript would allow the authors to make stronger conclusions regarding BSTOA as a probiotic.

Thank you very much for your valuable feedback about the pharyngeal muscle. The pharyngeal muscles could

also be stained with phalloidin, similar to the body wall muscles; however, due to the processing steps, such as fixation, it was unfortunately not possible to maintain their original shape as in the living state. Consequently, evaluations regarding aging or functionality could not be derived from the images of stained pharyngeal muscles. If given the opportunity, we would like to examine whether BSTOA can suppress morphological changes caused by the aging of pharyngeal muscles.

• Similar to the comment above regarding the visual assessment of actin bundles in body wall muscle, **the criteria that the authors used to evaluate the integrity of the intestine (Supp. Fig. 1b) should be spelled out in the Methods section.** Presumably the “rate of intestine” represents the condition of the intestinal tissue in adult animals as determined by a rubric that the authors converted to a numerical score, but the details of this procedure should be described. In addition, including representative micrographs to illustrate the condition of the intestine in 3 each group of animals would be a valuable complement to the micrographs of body wall muscle in Fig. 1f and should be strongly considered.

The area of the body and intestine of nematodes photographed under bright field microscopy (SZ2-ET, OLYMPUS) was measured using ImageJ by outlining the contours of each. A total of 40 worms were used per condition in three replicates.”

Thank you very much for your constructive advice. We have appended a revised description of the evaluation method for the percentage of intestines relative to the body in Lines 359–362 of the revised main text. Additionally, representative photographs for each group have been added.

“The percentage of the intestines in the body

The area of the body and intestine of nematodes photographed under bright-field microscopy (SZ2-ET, OLYMPUS) was measured using ImageJ by outlining the contours of each. A total of 40 worms were used per condition in three replicates.”

3. The inconsistencies between the measurements of gene expression by qRT-PCR as presented in Figs. 2a and 3c are troubling. In general the error bars on the average fold change values reported on the bar graph in Fig. 3c are substantially larger than the error bars for values reported in Fig. 2a. This is true even for some of the same genes whose expression was measured in both Figures (e.g. *gst-4* and *gst-38*). In some cases the range of the error bars exceeds the average fold change reported, calling the validity of the data into question. Can the authors account for the variability in fold values in Fig. 3c and why they did not observe this in Fig. 2a? Another inconsistency is that in Fig 2a, *gst-38* expression is increased in

BSTOA_day13 animals relative to OP50_day1, but in Fig. 3c its expression in BSTOA_day13 animals is lower than in OP50_day1 worms. To improve the quality of the gene expression data in Fig. 3c additional replicates of the qRT-PCR experiments are likely necessary and should be performed with RNA isolated from new cohorts of animals.

Thank you for your feedback. It is frequently observed that variations occur due to unpredictable factors such as the timing or season of experiments under different synchronized events. We referred to the qPCR data of the screening genes and focused on genes with expression changes of more than twofold. Our objective was to identify genes related to the maintenance of locomotion function and anti-aging effects in BSTOA, and to advance subsequent experiments using gene knockout models of the worm. Therefore, we emphasized not the variability between independent experiments, but rather the overall trends in gene expression. As pointed out, we conducted additional qPCR experiments and revised the data. Some genes exhibited a reduction in error bars, while others did not. Regarding the expression of *gst-38*, when comparing BSTOA_day13 to OP50_day1, we observed an increase in Fig. 2a and a decrease in Fig. 3c. In Fig. 3c, the average appeared to show a decrease, but the error bars extended toward the direction of an increase. This is likely due to individual variability, as mentioned earlier. On the other hand, when comparing BSTOA_day13 to OP50_day13, while there were slight differences in the fold change, both Fig. 2a and Fig. 3c consistently indicated a trend of increased expression in BSTOA_day13 compared to OP50_day13. Therefore, we concluded that these associated genes are involved remains valid.

4. In further regard to the gene expression data in Figs. 2a and 3c:

- The authors should make available a file that contains both the raw data from their qRT-PCR experiments along with the calculations that they performed to determine the fold change in expression.

As you pointed out, we have created the data file and attached it as Supplementary Table 3.

- Statistical tests should be performed to determine the significance of the changes in gene expression they observed between the different groups of animals tested, and the results should be reported in a supplemental table

Thank you for your constructive feedback. We conducted statistical tests and have presented the results in Figs. 2a and 3c, respectively.

- The nucleotide sequences of all primers used in qRT-PCR experiments should be made available in a supplemental table. If the authors used primers that have been published previously, then they should state this in the Methods section and provide the relevant literature references.

We have added a table of primer sequences and attached it as Supplementary Table 2. Thank you for pointing this out.

5. Analysis of the roles of *gpx-7* and *gpx-4* in modulating the effects of the BSTOA diet on older animals has the potential to provide mechanistic details to help to explain the authors' observations regarding the potential late-life health benefits of BSTOA as a nutrition source. Considering the significance of the data pertaining to these genes to the authors' story, I have concerns regarding the following: 1) the wording regarding the effect of BSTOA on *gpx-4* and *gpx-7* expression and the requirement for their expression for muscle functions in BSTOA animals, 2) the content and presentation of the functional characterization data in Fig. 4, and 3) the

lack of interpretation and contextualization of the results of the *gpx-4* and *-7* functional characterization experiments in the Discussion section. My specific comments on each of these points are presented below.

Wording regarding the expression and function of *gpx-4* and *gpx-7*

• In the Abstract the authors state that “consuming BSTOA may prevent the decline in age-related locomotor functions...through the suppression of *gpx-7* expression in aged worms” (lines 22-24, emphasis added). The qRT-PCR data in Fig. 3c shows that the expression levels of *gpx-7* are higher in Day 13 BSTOA-fed adults as compared to age-matched animals reared on OP50 or BS168. This suggests that the BSTOA diet maintains more youthful levels of *gpx-7* expression, curtailing the age-related decline in expression. **All statements describing the effect of BSTOA on *gpx-7* levels in adult worms throughout the manuscript should be edited to reflect this.**

As suggested, we have added descriptions to Lines 154–158 of the revised main text. Thank you very much.

• **It is critical for the authors to distinguish between the effects that the BSTOA diet has on expression levels of *gpx-4* and *gpx-7* in Day 13 animals.** Instead of describing their levels as “fluctuating” in BSTOA-fed adults, the authors should make clear that, as compared to when OP50 or BS168 are provided as nutrition sources, BSTOA causes a relative decrease in *gpx-4* levels and an increase in *gpx-7* levels.

As you pointed out, we have added additional descriptions to Lines 147–154 in the revised main text. Thank you very much for your valuable input.

• The authors’ characterization of *gpx-4* and *gpx-7* mutants led to several surprising results that are contrary to the predicted effects of loss-of-function mutations in those genes with regard to age-related phenotypes based on the changes in their expression levels brought about by the BSTOA diet. For example, while the absence of *gpx-4* yields no further benefit to BSTOA_Day 13 animals with regard to locomotion, pharyngeal pumping, or resistance to oxidative stress, it unexpectedly negates the effect of the BSTOA diet on the defecation cycle of older adults. This suggests that even though the BSTOA diet reduces *gpx-4* expression, *gpx-4* is nonetheless required for BSTOA to improve the adult defecation cycle. **The authors need to make this apparent by clarifying their description of the characterization of *gpx-4* mutants in the Results section.**

Content and presentation of the functional characterization data in Fig. 4

In their functional characterization of *gpx-4* and *gpx-7* mutants, the authors should strive for consistency between the data presented in Fig. 4 and the data in Fig. 1. Specifically, the following points should be addressed:

• **Assess movement velocity of *gpx-4* and *gpx-7* mutants at Days 1 and 13 of adulthood in BSTOA-fed animals as compared to age-matched wild type controls and OP50-fed animals**

We considered that the thrashing movement has more uniform evaluation conditions for assessing the function of body wall muscles than that of the moving velocity, as thrashing movement is a semi-forced activity, whereas moving velocity involves the nematode's voluntary movement. When evaluating moving velocity, it is difficult to differentiate whether reduced movement is due to aging, changes in the environment, or behavioral choices of the worm. Furthermore, it is challenging to eliminate the risk of injuring aged nematodes,

which often exhibit a lack of firmness and fragility during experimental handling. Therefore, we deemed that the thrashing movement assessment alone would suffice to evaluate the function of the body wall muscles. We hope this explanation meets your expectations.

- **in all panels, include data from all three genotypes reared on both diets at both Day 1 and Day 13 of adulthood.** As it stands the reader must refer to data from Fig. 1 to compare the effect of the *gpx-4* and *gpx-7* mutations to wild type Day 13 animals fed the BSTOA diet since data from that control is missing in panels A, C, and D. If these controls were not assessed alongside the mutants in the experiments for Fig. 4, then at a minimum the authors should create a table that summarizes the data from all behavioral assays (i.e. the data from Figs. 1 and 4) and allows the reader to make a direct comparison between different genotypes and treatment groups in one place.

Thank you for your suggestion. Regarding the Day 1 data for wild-type and *gpx-4* and *gpx-7* mutants, we have compiled the information in Supplementary Figure 3 to facilitate comparison. The data for Day 13 in each worm have been consolidated into a single figure, in Fig. 4. We hope these modifications meet your expectations.

- **The y-axis label in Fig. 4b should be changed to “defecation cycle” and the range of values should be adjusted to more closely match the range in Fig. 1a.**

The defecation intervals of aged worms are too prolonged, and the increase in group numbers in experiments using mutant strains rendered it practically difficult to investigate all groups in a single experiment. Therefore, the evacuation cycles analyzed in Fig. 1 were not examined in Fig. 4. Upon analyzing the data obtained in Fig. 1, we found that measuring the defecation time and investigating a sufficient number of worms was enough to determine differences in evacuation time. Consequently, for Fig. 4, we focused solely on measuring evacuation times and used these measurements as the data.

- **There is not an adequate explanation of what the values in the bar graph in Fig. 4c represent. Presumably these are survival percentages at a single time point in the oxidative stress assay with paraquat. To stay consistent with Fig. 3b, the authors should replace these data with a plot depicting the complete survival curves for all animals tested.** The current bar graph could be included in the supplement.

Thank you for your suggestion. To demonstrate consistency with Figure 3b, we thought the survival rate after two hours was sufficient and presented the data as a bar graph instead of a time-course analysis. Additionally, we added the following explanatory text to Figure 4c: “(c) The survival rate of various nematodes in paraquat assay after two hours”. We hope this explanation meets your expectations.

- **A second surprising result is that although mutations in *gpx-4* and *gpx-7* both affect body wall muscle in BSTOA-fed adults, they seem to be important for separate and discrete functions (defecation and locomotion, respectively). This suggests that they may not affect the totality of body wall muscle function during aging and may not preserve its structure to prevent sarcopenia. The authors should test this possibility by using fluorescence microscopy to examine actin microfilaments in BSTOA_Day13 *gpx-4* and *gpx-7* mutants. In addition, in the Discussion section the authors should address whether their results indicate that *gpx-4* and *gpx-7* could play roles in maintaining subsets of neuromuscular synapses or that their critical site of action may not be in muscle but instead in motor neurons.**

Thank you for your suggestion. We have evaluated the results of fluorescence microscopy observations of actin microfilaments in *gpx-4* and *gpx-7* mutants, injecting BSTOA for 13 days, and summarized them in Supplementary Fig. 3. The discussion has been included in the revised main text, Lines 243–249.

Lack of interpretation and contextualization of the results of the *gpx-4* and *-7* functional characterization experiments in the Discussion section

• **The Discussion would greatly benefit from a brief description of glutathione peroxidases and the gpx family of genes in *C. elegans*.** The authors should address the following questions: 1) What is the function of glutathione peroxidase enzymes and what role do they play in oxidative stress resistance?, 2) Why might two genes of the same family (i.e. *gpx-4* and *gpx-7*) be differentially regulated in the presence of BSTOA such that one is upregulated while the other is downregulated?, and 3) How do the authors findings regarding the function of *gpx-4* and *gpx-7* compare to previous functional characterization of gpx genes in *C. elegans*? As part of the answer this question, the authors should comment on Sakamoto et al., Genes to Cells (2014).

1) What is the function of glutathione peroxidase enzymes and what role do they play in oxidative stress resistance?

Thank you for your comment. I added to the revised main text. (Line 219–225)

2) Why might two genes of the same family (i.e. *gpx-4* and *gpx-7*) be differentially regulated in the presence of BSTOA such that one is upregulated while the other is downregulated?

Thank you for your comment. Although it remains within the realm of speculation, *gpx-7*, which is primarily expressed in the gut, may have exhibited higher expression levels in BSTOA_day13 compared to OP50_day13 and BS168_day13, likely due to better gut integrity in BSTOA as shown in Supplementary Fig. 1. Regarding *gpx-4*, it is possible that the reduced stress caused by ROS in BSTOA_day13 might have resulted in lower gene expression in BSTOA_day13 compared to OP50_day13 and BS168_day13. However, we have not investigated that experimentally. Unfortunately, definitive conclusions cannot be drawn.

3) How do the authors findings regarding the function of *gpx-4* and *gpx-7* compare to previous functional characterization of gpx genes in *C. elegans*?

I have added comments regarding this matter to the revised main text of the Discussion section, specifically in Lines 227–236.

• In addition to those mentioned above, two other unexpected results uncovered through functional characterization of *gpx-4* and *gpx-7* mutants should be addressed. First, in OP50-fed adult animals, mutations in *gpx-4* fail to ameliorate the behavioral decline and enhanced sensitivity to oxidative stress. At the same time, mutations in *gpx-7* do not exacerbate those same phenotypes. **The authors should address this in the Discussion, perhaps as part of a broader consideration of ROS detoxification strategies (see below).** Second, despite their predicted role in detoxifying oxidants, neither *gpx-4* and *gpx-7* appear to be required for oxidative stress resistance of Day 13 animals, regardless of their nutrition source.

The authors should comment on the following points: What does this say about the redundancy of ROS

detoxification strategies in *C. elegans*, and does it suggest a possible hierarchy among them in older animals? If *gpx-4* and *gpx-7* primarily function in muscle (and/or neurons) but are not required to protect the entire animal from oxidative stress, could there be tissue-specific modes of 6 counteracting oxidative stress? Is this possibility supported by other studies in the literature from *C. elegans* or other species?

Thank you for your comment. It has been elucidated that the antioxidant gene network in *C. elegans* does not rely solely on a single ROS elimination mechanism but rather consists of multiple genes forming interconnected and redundant networks that adaptively activate under stress conditions. Furthermore, this redundancy has been noted to exert complex influences on both organismal lifespan and stress resistance. Additionally, it has been suggested that localized expression patterns of oxidative stress-related genes may play crucial roles in specific tissues. (<https://doi.org/10.1155/2012/608478>)

6. The authors are strongly encouraged to revise and edit their manuscript to adjust word choice and phrasing to make their statements more clear. In addition they should provide some missing information, as described below.

General comments regarding writing

- The authors are strongly encouraged to develop a more cohesive narrative throughout the manuscript. This involves providing readers with greater insight into their rationale for experiments, describing expected results, and highlighting particularly important results that changed the direction of the project or shaped the conclusions. Specifically with regard to the experiments where gene expression levels were measured (Figs. 2a and 3c), the authors should: 1) provide further detail about the specific genes studied (i.e. functional categories they represent, associated GO terms, etc.), 2) explain the expression patterns that the authors were looking for to identify genes of interest—this is especially important to articulate for *gpx-4* and *gpx-7* in Fig. 3c.

- 1) provide further detail about the specific genes studied (i.e. functional categories they represent, associated GO terms, etc.),

I have prepared supplementary materials.

- 2) explain the expression patterns that the authors were looking for to identify genes of interest—this is especially important to articulate for *gpx-4* and *gpx-7* in Fig. 3c.

Thank you for your comment. Additions have been made to Lines 147–158 of the revised main text.

Introduction section

- While the influence of probiotics on diverse tissues in vertebrate animals with circulatory systems and more extensively innervated organs is more familiar, in the Introduction section the authors should briefly address the prevailing model in the *C. elegans* field for how ingesting probiotic strains of bacteria affects organismal physiology in the comparatively simpler invertebrate system. In addition, they should point out specific data in reference 7 that support their statement that “probiotics represent a promising strategy for preventing sarcopenia” (line 44).

Thank you for your suggestion. We have added a brief mention of the general model commonly used in the nematode field regarding the impact of bacterial probiotic strain intake on physiological functions in

relatively simple invertebrate systems in the revised main text, Lines 53–56. Additionally, for Reference 7 stating that “Probiotics are a promising strategy for preventing sarcopenia” (Line 44), we have supplemented this with more specific data in the revised main text, Lines 43–47.

- The authors’ description of the Koyuncu *et al* study (Ref 20) needs to be clarified. They mention that “knockdown of the gene that is upregulated with aging in worms extended lifespan” (Lines 56-57). There are widespread age-related changes in gene expression in *C. elegans*. Are the authors meaning to refer to all of the genes that are upregulated during aging or to a specific one? If it is only one gene, which is it and why is it significant? How is it related to *B. subtilis* as a food source for the worms?

Thank you for your feedback. Since it referred to the specific gene *eps-8*, I have added a note to Line 62 in the revised main text.

Methods section

- Additional methodological details regarding image acquisition and processing should be provided. In particular, at what total magnification were the images collected? Were all of the images collected with the same camera settings (e.g. exposure time)? What type of image processing, if any, was performed on the raw data and what software was used?

Thank you very much for your advice. I have added detailed methods in the revised main text, Lines 330–335.

- The methods used to measure worm body size as reported in Sup. Fig. 1a need to be described. In that figure the authors refer to “body size” in “bits”. This should be converted to “body length” in microns.

The method for measuring nematode body length reported in Supplementary Figure 1a has been converted to "body length" in microns.

Minor concerns

1. In the title of the manuscript the authors should consider including a reference to the effect of the BSTOA diet on oxidative stress resistance in adult animals.

In this study, experiments related to oxidative stress have been conducted only in the aspect of paraquat assay. Furthermore, while oxidative stress-related genes are utilized as genes associated with the anti-aging of locomotion functions, the detailed roles of *gpx-4* and *gpx-7* in *C. elegans* have only been reported as inferred characteristics. Therefore, referring to oxidative stress resistance in the title might lead to misunderstandings. Following the advice received, we have carefully discussed the title again, but would like to avoid making any changes. Thank you very much for your valuable comment.

2. The title to Fig. 1 does not apply to the micrograph in panel (f) and associated bar graph in panel (e). The authors should either edit the figure title or consider converting panels (e) and (f) to a separate stand-alone figure.

The title of Figure 1 has been corrected. Thank you very much for pointing it out.

3. The account of the results of the qRT-PCR studies to measure gene expression in OP50_{day13} and

BSTOA_day13 animals needs to use clearer language and be more descriptive. Instead of grouping all genes whose expression levels in OP50_day13 and BSTOA_day13 animals differ from their levels in OP50_day1 animals by 2-fold or more, separate these into different sets according to whether or not they are up- or downregulated in BSTOA_day13 worms. This would help to bin genes into different categories, potentially revealing commonalities between them, and it would remove the ambiguity associated with the term “fluctuated” (line 102). Also, the heatmap in Fig. 2a is a valuable representation of genes with similar differences in expression between the two groups of animals and could be used to guide the authors’ description of their qRT-PCR experiments in the Results section.

As per your suggestion, we have revised the main text as shown in Line 110–116 and 147–158.

4. Figs. 2 and 3—Since the y-axis on the plot in panels 2a and 3c represent Log₂ fold change, there is no reason to express the values as exponentials. Instead the range should be from -5 to 7. The same comment applies to the notation on the heat maps in both figures.

As per your comments, we have made the necessary adjustments. Kindly review the revised versions of Fig. 2a, 2b, and Fig. 3c, d.

5. There are many errors throughout the text that should be corrected. Representative examples are presented below.

Grammatical errors

- line 28—either need a semicolon in this sentence after “countries” or need to begin another sentence with “However”

As noted, the “,” in Line 28 of the revised text has been corrected to a “;”.

Editorial mistakes

- Direct quotes are used in lines 39 and 40

To clearly indicate the direct citation, “by the Food and Agriculture Organization of the United Nations and the World Health Organization” has been added to Lines 39–40 of the revised main text.

- Repeated words in line 66

The duplicate term “to elucidate” has been removed.

- Wording in lines 127 and 128 is awkward and should be revised: “Notably function differences were noted...”

We revised the text in Lines 137–139 of the manuscript from “Notably function differences were noted...” to “Functional differences were clearly observed among *B. subtilis* strains, indicating that the suppression of locomotor dysfunction and enhancement of oxidative stress resistance would be characteristic of BSTOA.”

- The phrase “which changes characteristically with the suppressed decline in locomotor function” in line 131 is unclear. Are the authors referring to oxidative stress resistance as the phenotype that changes? Also, what is meant by the “suppressed decline in motor function”? Does this refer to the effect of the BSTOA

diet on adult animals?

It refers to the "suppression of decline in motor function." This was indicated by referencing Fig. 3a.

- Line 190: should be changed to “BSTOA has effects both on longevity and the maintenance of...”

The related sentence has been revised as follows: “Therefore, this is the first study to report that BSTOA has effects on both lifespan extension and the maintenance of locomotor function through suppressing the age-related decline in *gpx-7* gene expression.”

- The sentence that begins “Among the many genes” on line 193 needs to be reorganized and edited for clarity.

The related sentence was revised as follows: “Among the many genes related to aging and movement, the expression of genes associated with oxidative stress was up-regulated more than twice after examining the expression of several representative genes by qPCR (Fig. 2).”

- Line 240 reads “removing t56he agar”

As shown in Line 267 of the revised main text, the typographical error "56" in "removing t56he agar" has been deleted.

Spelling errors

- Line 278: do the authors mean to write “thrash” instead of “slash”

I corrected the misspelled word "slash" to "thrash" as seen in Line 307 of the revised main text.

Re: COMMSBIO-24-7847A

Title: *Bacillus subtilis* TO-A suppresses age-related decline in locomotion of *Caenorhabditis elegans*

We are grateful to the reviewers for their careful assessment of our paper. Guided by their constructive feedback, we have re-revised the manuscript and refined the text where needed. Below, we present a detailed, point-by-point reply to every comment raised.

Response to Reviewer 1:

The revised manuscript titled "*Bacillus subtilis* TO-A suppresses age-related decline in locomotion of *Caenorhabditis elegans*" offers a well-executed and insightful investigation into the protective effects of *B. subtilis* TO-A against age-related physiological decline in *C. elegans*. The authors have carefully addressed all previous reviewer comments, resulting in substantial improvements to the manuscript's clarity, scientific rigor, and overall quality. The presented data convincingly demonstrate that *B. subtilis* TO-A alleviates the age-associated deterioration in locomotor function in *C. elegans*. Revisions to the methodology section have enhanced transparency, and the figures and legends are now more informative and accessible. Furthermore, the statistical analyses are appropriate and well-applied. Given the comprehensive nature of the revisions and the robustness of the findings, I find the manuscript to be suitable for publication and recommend acceptance in its current form.

Thank you very much for your encouraging comments; they are truly motivating for our future research. We are sincerely grateful for your thoughtful feedback.

Response to Reviewer 2:

Manuscript Number: COMMSBIO-24-7847A

Bacillus subtilis TO-A suppresses age-related decline in locomotion of *Caenorhabditis elegans*

The reviewer pleased that the authors have resolved all the questions. However, based on your response, there are three points that should be clarified.

We would like to express our sincere gratitude for your careful evaluation of our manuscript and for your helpful comments and advice. Your thoughtful comments and suggestions have been invaluable in helping us to improve the quality of our work.

1. Regarding *gpx-4* and *gpx-7*

The involvement of these genes as you describe in your response letter should be validated using mutants lacking both or transgenics with reduced expression by RNAi.

Thank you for your valuable feedback. Our qPCR analysis showed that *gpx-4* expression was increased and *gpx-7* expression was decreased in aged worms. Furthermore, in BSTOA_day13, the expression level of *gpx-4* was comparable to that observed in OP50_day1, and the age-related decrease in *gpx-7* expression was mitigated. These results suggested that *gpx-4* may contribute to the maintenance of defecation behavior during aging, while *gpx-7* suppresses the age-associated decline of thrashing behavior.

As the aim of this oxidative stress gene analysis was to clarify the contribution of individual genes, we did not generate double mutants lacking both *gpx-4* and *gpx-7*. Additionally, since *gpx-4* and *gpx-7* are predicted to be localized in different tissues, these mutants may result in more complex and difficult-to-interpret phenotypes. In future studies, we plan to expand the scope of our analysis, generate double mutants as necessary, and investigate whether BSTOA exerts its effects at multiple levels. Your comments have been very constructive and will be in guiding for our future research.

2. If the gene examined by qPCR is the major gene reported so far, the literature should be cited where the gene is mentioned.

Thank you for pointing this out. We have cited references 21 and 25–33 on Line 107 and added them to the appropriate positions in the References section of the re-revised version.

3. About the housekeeping used in qPCR

I understood that 5 genes were used as housekeeping to minimize the effect in aging. But how did you correct the expression levels for these five genes? Did you just simply graph the average of the 5 data? Please describe in detail how you calculated and graphed the data in the Methods section.

Thank you for your suggestion. We have added the information in the Materials and Methods section of the re-revised version (Lines 346–347, 348, 349, and 353–354) as follows:

“Each gene was measured in triplicate and its Ct value was averaged across three wells; the obtained data were then normalized to the average Ct value of two (*tba-1* and *cdc-42* [Fig. 2]), or five genes (*tba-1*, *cdc-42*, *eif-3*, *ama-1* and *pmp-3* [Fig. 3]) as the reference⁵³ within each experiment, and the designed primers are listed in Supplementally Table 1. We used the delta-delta threshold cycle (Ct) method ($2^{-\Delta\Delta C_t}$) to express the difference between attention groups. Delta Ct was generated by subtracting the average Ct of the reference genes from the Ct of each gene of interest. Then, the $\Delta\Delta C_t$ was derived from comparison with a control sample. The $\Delta\Delta C_t$ results were plotted on graphs using a logarithmic axis (Fig. 2a and Fig. 3c).”

Inomata and colleagues have revised their manuscript describing the effect of a probiotic bacterium (referred to as BSTOA) on the structure and function of muscle in older adult *C. elegans*. In their rebuttal letter, the authors systematically explain their responses to nearly all of the critiques that I described in my initial review of the original version of the manuscript. I appreciate that the authors revisited some experiments, generated new data and analyses that appear in both the main and supplemental figures, and added content to the Discussion section. Further, they have added missing details of their methodology and reworded some descriptions of their data, as I requested. In my opinion these changes enhance the manuscript by improving the quality of some of the data while at the same time uncovering new evidence that impacts the conclusions that can be drawn from the study as a whole. At this point, additional experiments are not necessary. However, I do not believe that the authors have gone far enough in their analysis and writing to present a clear picture of their results and conclusions. I feel that further revisions of the manuscript are necessary in order for it to meet the standards of the journal.

I appreciate that the authors addressed many of my comments regarding their writing. There are still several issues that need to be corrected, though. For example, the authors did not change the wording of the abstract regarding the expression of *gpx-7* as I requested. I also believe that they should revisit their word choice in describing some of their results to make their points more clear to readers. Finally, I found many typographical and grammatical errors throughout the revised version of the manuscript (text and figures).

The major concerns that I articulated in my original review of the manuscript can be summarized according to three main points, which I present below along with new commentary on how the authors addressed them in the revised version of their manuscript.

1) The authors were asked to provide clearer descriptions of age-related phenotypes with respect to deterioration of muscle and intestinal structure and how this is affected by the BSTOA diet.

- I requested that the authors add new micrograph of body wall muscle from OP50_day 13 and BSTOA_day 13 animals that are taken at the same magnification, focus on the same region of the animals' body, and are free of saturated pixels to better illustrate the structure of body wall muscles. The authors adjusted the brightness of their original image of OP50_day 13 animals in Fig. 1f and added red arrowheads to draw attention to the muscle cells. They replaced their original micrograph of BSTOA_day 13 animals with a new one that has more uniform signal without any apparent oversaturation. These two changes help to make the muscle

structure phenotypes more clear and allow for more straightforward comparison between images.

- I requested that the authors expand their analysis of muscle tissue architecture in BSTOA-fed animals to include pharyngeal muscle. The authors made a good-faith effort to examine pharyngeal muscle structure in OP50-Day 1 and Day 13 animals through phalloidin staining. This did not, however, lead to interpretable data because the fixation and processing steps involved in the staining procedure altered the shape of the pharynx. The authors could have pursued an alternative approach to acquire these data (such as through imaging a strain that expresses a GFP or RFP-tagged MYO-2), which may have allowed them to make a more accurate conclusion regarding the effect of the BSTOA diet on overall muscle integrity in older animals. In my opinion more data are necessary in order to draw broader conclusions about how BSTOA affects other types of muscle, but with the data in hand the authors can justify a claim that BSTOA maintains the structure of *body wall muscles* in older adults, which is consistent with their functional data.

- I requested including representative micrographs to illustrate the condition of the intestine in OP50_Day 13 and BSTOA_Day13 animals. The authors have added these images to Supp. Fig. 2c to accompany their quantitative assessment of intestinal structure in panel b of that same figure, and they briefly describe the methodology used to determine the reported ratios. I reiterate my previous comment that the y-axis label of “rate of intestine” is unclear and should be revised. Regardless, these new micrographs help to more completely describe the effects of the BSTOA diet on tissue architecture during aging.

Although minor writing revisions are still required, I am satisfied with the authors’ response to this concern.

2) The authors were asked to improve the quality of data reporting on the effect of the BSTOA diet of gene expression by including additional qRT-PCR experiments with the goal of achieving better reproducibility between biological replicates.

- The authors conducted additional qRT-PCR experiments that yielded data that reduced the standard error of the mean for the reported expression levels of some genes. This was particularly helpful in the case for certain genes where, in the data shown in the original version of the manuscript, the range of the error exceeded the average fold change in expression (e.g. *gpx-5*, *gst-4*, and *gst-38*). In general, the new qRT-PCR results also make the data for genes whose expression is reported in both

Figs. 2a and 3c more consistent between the figures. (Note: The qRT-PCR data in Fig. 3 of the revised manuscript is lacking a label denoting it as panel “c”.)

- Although there is some discrepancy between the reported expression levels of *gst-38* in Fig. 2 versus Fig. 3 when comparing expression levels in OP50_day 1 animals and BSTOA_day 13 animals, the authors argue that this may be attributed to “individual variability” and that they observed an overall trend of increased *gst-38* expression in BSTOA_day 13 animals. I concede this point to the authors.
- More importantly, the data for *gst-4* expression in Fig. 3c is substantially improved, with a significantly reduced standard error.

Overall the qRT-PCR data in the revised version of the manuscript appear to be more reproducible between replicates as compared to the data in the original version. This helps to strengthen the validity of their claims regarding the transcriptional response of animals to the BSTOA diet.

3) The authors were asked to clarify the functional roles of *gpx-4* and *gpx-7* in preserving tissue structure and behavioral functions during aging.

- I requested that the authors assess the movement velocity of BSTOA-fed *gpx-4* and *gpx-7* mutants at Days 1 and 13 of adulthood. In their rebuttal, the authors made a compelling argument for choosing to evaluate thrashing instead of movement velocity. Part of their rationale is that as compared to a “semi-forced” behavior (thrashing), fluctuations in voluntary movements can be influenced by a number of variables that make its interpretation complicated.
- I requested that the authors examine actin microfilaments in BSTOA-Day 13 *gpx-4* and *gpx-7* mutants to determine whether these genes are important for preserving the structure of all body wall muscles. The authors performed these experiments with the appropriate controls and have included new data summarizing the results from manual scoring of muscle structure by fluorescence microscopy in Supp. Fig. 3. In my opinion, these data are critical and they affect the conclusions that can be drawn from the work as a whole when taken together with other evidence reported in the manuscript.

Additional comments on this point are below.

• I asked the authors to contextualize results regarding gpx-4 and gpx-7, especially with regard to their role in oxidative stress resistance, and to expand the interpretation of their data. In the revised manuscript, the authors included a full paragraph in the Discussion section that invokes data from other studies that speak the apparent functional redundancy of gpx family genes and to a potential role for gpx-4 and gpx-7 in regulating lipid oxidation. By relating their data to other studies and offering an explanation as to why gpx-4 and gpx-7 mutants show no enhanced sensitivity to oxidative stress, the authors have improved the quality of their Discussion. However, I disagree with some statements that the authors make about their data, and I do not feel that they have adequately synthesized the totality of the evidence that they report to offer the reader a clear and succinct account of how (or if) gpx-4 and gpx-7 underlie the mechanistic basis for the improved muscle structure and motor function in BSTOA-fed adult animals.

The relative changes in expression of gpx-4 and gpx-7 in BSTOA_day 13 animals lead to predictions about how the two genes affect muscle structure and function during normal aging in animals fed the standard OP50 diet and about what role they may play in facilitating the beneficial effects of the BSTOA diet. In particular, the reduced expression of gpx-4 in BSTOA-fed adult *C. elegans* suggests that it may be detrimental to muscle structure and function later in life and that it could have no contribution to the benefits derived from BSTOA. On the other hand, because gpx-7 expression is elevated in BSTOA-fed animals, it may be expected that age-related muscle decline is exacerbated in gpx-7 mutants and that gpx-7 plays a key role in mediating the effects of BSTOA. A major sticking point in the narrative of the manuscript, in my opinion, is that almost none of these predictions turn out to be correct, and the authors do not provide an explanation as to why. Other than citing the potential functional redundancy between gpx family genes, can the authors reconcile the expression data from wildtype BSTOA-fed adults with the behavioral data of gpx-4 and gpx-7 mutants?

The authors need to synthesize data from multiple lines of evidence to formulate more direct statements regarding what their data reveal about the function of gpx-4 and gpx-7. Taking a holistic view of the authors' complete dataset, I believe that the following points should be made in clear language in the manuscript:

- A) During aging in animals fed the standard OP50 diet, gpx-4 is required to preserve body wall muscle structure (Supp. Fig. 3), and it plays a small role in maintaining thrashing and defecation behaviors (Fig. 4a, b).

Despite the decrease in expression brought about by the BSTOA diet (Fig. 3c), gpx-4 must be maintained above threshold levels in order for aged animals to gain the full benefit of BSTOA in preserving muscle structure (Supp. Fig. 3) and motor function (both thrashing and defecation; Fig. 4a, b). Therefore, regardless of nutrition source, gpx-4 is important for maintaining the structure of body wall muscle during aging, and this may directly contribute to upholding behaviors that require body wall muscle contraction.

B) gpx-7 may be detrimental to muscle structure during aging (Supp. Fig. 3) and is not important for maintaining thrashing or defecation behaviors later in life in animals reared on OP50 (Fig. 4a, b). It plays no role in preserving muscle structure in BSTOA-fed animals (Supp. Fig. 3), nor is it required for the improved evacuation cycle of older animals on the BSTOA diet (Fig. 4b). However, it is important for the enhanced thrashing associated with BSTOA in aged adults (Fig. 4a), consistent with predictions based on its elevated expression in those animals. As compared to gpx-4, gpx-7 seems to have a more limited role in motor function during aging that is unlikely to contribute to maintaining muscle structure and may be specific to only certain behaviors in the context of discrete nutritional conditions.

With regard to (A), the authors' statement in the Results section that "BSTOA ingestion might be associated with gpx-4 expression, but not gpx-7, in the defecation function of aged worms..." (Lines 177-178) could be interpreted to mean that BSTOA *induces* expression of gpx-4 when, in fact, it lowers the expression of gpx-4 in Day 13 animals as compared to OP50-fed controls (but does not completely eliminate gpx-4 expression). It should be edited to accurately reflect the facts.

Further, the statistical analyses in Fig. 4a and the description of the thrashing phenotypes of BSTOA-fed day 13 gpx-4 and gpx-7 mutants in the Results section (lines 165-167) suggests that both of these genes are important for BSTOA to enhance locomotion in aged adults. I agree with this interpretation of the data. However, the authors state that "[t]hese results suggest that BSTOA specifically affects the expression of gpx-7 during the thrashing motion..." (Lines 169-170). This could be misconstrued by readers to suggest that gpx-7 is upregulated *only during muscle contractions*. More importantly,

it could be taken to mean that gpx-7 plays a bigger role than gpx-4 for maintaining thrashing behavior in BSTOA-fed adults when this is not supported by the data (Fig. 4a). The description of these results must be clarified.

I believe that the authors should carefully consider their results in aggregate and use more descriptive and precise language to describe the roles of gpx-4 and gpx-7 in preserving muscle structure and motor function during aging in a manner that is consistent with all of the evidence that they have uncovered in their studies. I recommend using points (A) and (B) above as a guideline. This is a critical element of their manuscript that must be included.